# INFERENCE OF UNOBSERVED EVENT STREAMS WITH NEURAL HAWKES PARTICLE SMOOTHING

## ABSTRACT

Events that we observe in the world may be caused by other, *unobserved* events. We consider sequences of discrete events in continuous time. When only some of the events are observed, we propose **particle smoothing** to infer the missing events. Particle smoothing is an extension of particle filtering in which proposed events are conditioned on the future as well as the past. For our setting, we develop a novel proposal distribution that is a type of continuous-time bidirectional LSTM. We use the sampled particles in an approximate **minimum Bayes risk decoder** that outputs a single low-risk prediction of the missing events. We experiment in multiple synthetic and real domains, modeling the complete sequences in each domain with a neural Hawkes process (Mei & Eisner, 2017). On held-out incomplete sequences, our method is effective at inferring the ground-truth unobserved events. In particular, particle smoothing consistently improves upon particle filtering, showing the benefit of training a bidirectional proposal distribution.

## 1 INTRODUCTION

**Event streams**, i.e., discrete events in continuous time, are often *partially* observed in the world. Given trained models of the complete data and the missingness mechanism, one can probabilistically predict the *unobserved* events, no matter whether they are **missing at random (MAR)** or **missing not at random (MNAR)**. Such an ability is useful in many applied domains:

- *Scientific experiments.* Scientific findings rely heavily on experimental observations, but due to practical limitations, measurements are often incomplete. For example, a systems neuroscientist will miss all the neural spikes from unmonitored neurons, but these are imputable to some degree from spikes observed elsewhere in the system, providing a more complete picture of neural activity.

- *Medical records.* Some patients keep a diary or use a smartphone app to log their behavior in between their hospital visits, but other patients don't. Out-of-hospital events such as symptoms, self-administered medications, diet, and sleep may correlate with monitored in-hospital events such as tests, diagnoses, and treatments. Inferring these events when they are not recorded may help doctors counsel patients.

- *Competitive games.* In competitive games such as StarCraft and poker, a player does not have full information about what her opponents have done (e.g., build mines and train soldiers) or acquired (e.g., certain cards). Correctly imputing what has happened "what I did" and "what I saw he did" would help the player make good decisions. Similar remarks apply to practical scenarios (e.g., military) where multiple actors compete and/or cooperate.

- *User interface interactions.* Many important user actions are not observed by the software. For example, users of an online news provider may have known what they need by reading an information-rich headline without clicking (to read more details). Such events are expensive to observe (e.g. via a gaze tracker) thus often missing. However, imputing them (to some extent) given the observed events (e.g. clicks) would be useful, e.g., helping the software to improve the user satisfaction in the long run.

- Other partially observed event streams arise in *online shopping*, *social media*, etc.

A flexible probabilistic model for complete event streams is the **neural Hawkes process** (Mei & Eisner, 2017), a recurrent neural model that allows past events to have complex influences on the

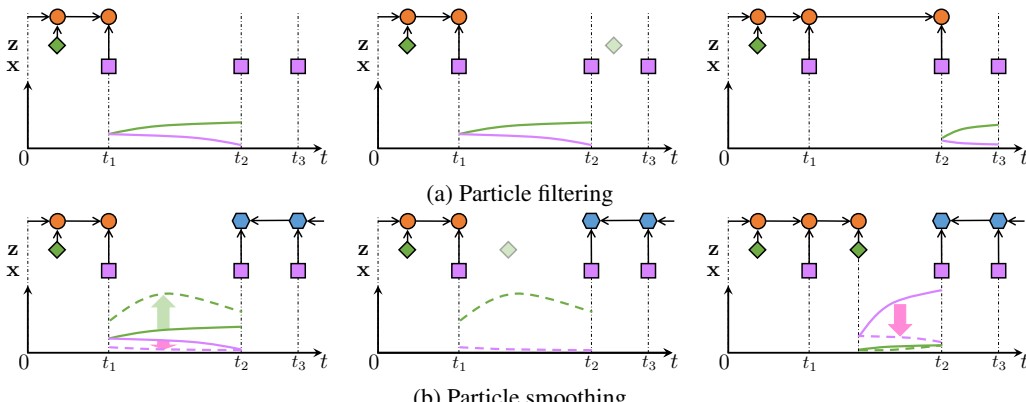

Figure 1: Stochastically imputing a taxi's pick-up events (◆) given its observed drop-off events (■). At this stage of the computation, we are trying to determine the next event after the ■ at time $t_1$.

In **Figure 1a (particle filtering)**, the neural Hawkes process's LSTM has already read the proposed and observed events at times $\leq t_1$. Its resulting state ● determines the model intensities ── and ── of the two event types (**left image**). Both intensities are low because there are few passengers at this time of day, so it happens that no event is proposed in $(t_1, t_2)$. Specifically, Algorithm 1 determines that the next proposed event (◆ in **middle image**) would be *somewhere* after $t_2$, though without bothering to determine its time precisely (line 36). Thus, ◆ is discarded (line 37), having been preempted by the observed ■@$t_2$. We continue to extend the particle by feeding ■@$t_2$ into the LSTM (**right image**) and proposing subsequent events based on the new intensities after $t_2$. But because ── was low at $t_2$, the ■@$t_2$ was unexpected, which results in downweighting the particle (line 28): intuitively, we have just realized that this particle will be improbable under the posterior, because its complete sequence will include consecutive drop-offs far apart in time (■@$t_1$, ■@$t_2$).

**Figure 1b (particle smoothing)** samples from a better-informed proposal distribution. A second LSTM (Appendix B) reads the future observations from right to left. Its state ● is used *together* with ● to determine the proposal intensities ∙∙∙ and ∙∙∙ (**left image**). Since a drop-off at $t_2$ strongly suggests a pick-up before $t_2$, considering the future increases the intensity of pick-up on $(t_1, t_2)$ from ── to ∙∙∙ (while decreasing that of drop-off from ── to ∙∙∙). Consequently, the next proposed event is more likely to be a pick-up in $(t_1, t_2)$ than it was in Figure 1a. If so (**middle image**), this event ◆@$t_{1,1}$ is fed into the neural Hawkes process's LSTM (**right image**). The updated state ● determines the new *model* intensities ── and ──: the higher ── intensity says that a dropoff is now expected, so we will not downweight the particle as much if $t_2$ is the next event. The updated ● also combines with ● to determine the new *proposal* intensities ∙∙∙ and ∙∙∙, which are used to sample the next event (either an unobserved event at $t_{1,2} \in (t_1, t_2)$, or the next observed event $t_2$).

type and timing of subsequent events. Suppose we have a trained version of this model, as well as a separate probability model (the "missingness mechanism") that stochastically determines which of the events will be observed. We can then easily use Bayes' Theorem to define the posterior distribution over the complete sequence $\mathbf{x} \sqcup \mathbf{z}$ given just the observed events $\mathbf{x}$: see equation (3) below. However, it is computationally difficult to reason about this posterior distribution. In this paper we provide approximate methods for sampling complete sequences from the posterior. In the remainder of the introduction, we provide a high-level sketch of these methods.

Mei & Eisner (2017) give an algorithm to sample a complete sequence from a neural Hawkes process. Each event in turn is sampled given the complete history of previous events. However, this algorithm only samples from the prior. We will adapt it into a **particle filtering** algorithm that samples from the posterior, i.e., given all the observed events. The basic idea (Figure 1a) is to draw the events in sequence as before, but now we *force* any observed events to be "drawn" at the appropriate times. That is, we add the observed events to the sequence as they happen (and they duly affect the distribution of subsequent events). There is an associated cost: if we are forced to draw an observed event that is *improbable* given its past history, we must downweight the resulting complete sequence accordingly, because evidently this particular past history—as determined by the previous draws— was inconsistent with the observed event, and hence cannot be part of a high-likelihood complete sequence. Using this method, we sample many sequences (or **particles**) of different *relative* weights.

Alas, this approach is computationally inefficient. Sampling a complete sequence that is actually probable under the posterior requires great luck, as the thinning algorithm must have the good fortune to draw only events that happen to be consistent with future observations. Such lucky particles would appropriately get a high weight relative to other particles. The problem is that we will rarely

get such particles at all (unless we sample a very large number of particles). To get a more accurate picture of the posterior, we can improve the method by drawing each event from a smarter distribution that is explicitly conditioned on the future observations (rather than drawing the event in ignorance of the future and then downweighting the particle if the future does not turn out as hoped).

This idea is called **particle smoothing** (Doucet & Johansen, 2009). How does it work in our setting? The neural Hawkes process defines the distribution of the next event using the state of a **continuous-time LSTM** that has read the past history from left-to-right. When sampling a proposed event, we now use an modified distribution (Figure 1b) that *also* considers the state of a second continuous-time LSTM that has read the future observations from right-to-left. As this augmented distribution is imperfect—merely a proposal distribution—we still have to reweight our particles to match the actual posterior under the model. But this reweighting is not as drastic as for particle filtering, because the new proposal distribution was constructed and trained to resemble the actual posterior.

Bidirectional recurrent neural networks have proven effective at predicting linguistic words and their properties given their left *and right* contexts (Graves et al., 2013; Bahdanau et al., 2015; Peters et al., 2018): in particular, Lin & Eisner (2018) recently applied them to particle smoothing for sequence tagging, To our knowledge, however, this is the first time such an architecture has been extended to predict events in continuous time.

## 2  PRELIMINARIES[1]

We consider a missing-data setting (Little & Rubin, 1987). We are given a fixed time interval $[0, T]$ over which events can be observed. Each possible outcome in our probability distributions is a **complete event sequence** in which each event is designated as either **observed** or **missing**. The random variables $Obs$, $Miss$, and $Comp$ refer respectively to the sets of observed events, missing events, and all events over $[0, T]$. Thus $Comp = Obs \sqcup Miss$, where $\sqcup$ denotes disjoint union. Under the probability distributions we will consider, $|Comp|$ is almost surely finite.

We will observe $Obs$ to be some particular set of events $\mathbf{x} = \{k_0@t_0, k_1@t_1, k_2@t_2, \ldots, k_I@t_I, k_{I+1}@t_{I+1}\}$, where each $k_i \in \{1, 2, \ldots, K\}$ is an event type and $0 = t_0 < t_1 < t_2 < \ldots < t_I < t_{I+1} = T$ are the times of occurrence in increasing order. This sequence always includes the special boundary events $k_0@t_0 = \text{BOS}@0$ ("beginning of sequence") and $k_{I+1}@t_{I+1} = \text{EOS}@T$ ("end of sequence").

Following each observed event $k_i@t_i$ for $0 \leq i \leq I$, we may hypothesize some $J_i \geq 0$ *unobserved* events $\{k_{i,1}@t_{i,1}, k_{i,2}@t_{i,2}, \ldots, k_{i,J_i}@t_{i,J_i}\}$, where $t_i < t_{i,1} < t_{i,2} < \ldots < t_{i,J_i} < t_{i+1}$.

We may regard the subscript $i$ as a shorthand for $(i, 0)$, so the observed events are $\mathbf{x} = \{k_{i,j}@t_{i,j} : j = 0\}$ and the hypothesized unobserved (missing) events are $\mathbf{z} = \{k_{i,j}@t_{i,j} : j \neq 0\}$.

The hypothesized *complete event stream* $\mathbf{x} \sqcup \mathbf{z}$ is thus indexed by pairs $\ell = \langle i, j \rangle$ ordered lexicographically, where $t_{\ell'} < t_\ell$ if $\ell' < \ell$. We write $\ell + 1$ for the index of the event immediately after $\ell$, so $\langle i, j \rangle + 1 \overset{\text{def}}{=} \langle i, j+1 \rangle$ when $j < J_i$ and $\langle i, j \rangle + 1 \overset{\text{def}}{=} \langle i+1, 0 \rangle$ otherwise. Each event $k_\ell@t_\ell$ (whether observed or not) may be influenced by the **history** $\mathcal{H}(t_\ell)$—the set of all *observed* and *unobserved* events that precede it, where we define $\mathcal{H}(t) \overset{\text{def}}{=} \{k_\ell@t_\ell : t_\ell < t\}$ for any $t \in [0, T]$.

We assume that the complete sequence $Comp$ was first generated by a complete data model $p$, and then some of the events were censored (marked as missing) by a **missingness mechanism** $p_{\text{miss}}$:

$$p(Obs = \mathbf{x}, Miss = \mathbf{z}) = p(Comp = \mathbf{x} \sqcup \mathbf{z}) \cdot p_{\text{miss}}(Miss = \mathbf{z} \mid Comp = \mathbf{x} \sqcup \mathbf{z}) \quad (1)$$

Furthermore, our complete data model (such as a neural Hawkes process) will take the factored form

$$p(Comp = \mathbf{x} \sqcup \mathbf{z}) = \Big( \prod_{i=0}^{I} \prod_{j=0}^{J_i} p(k_{i,j}@t_{i,j} \mid \mathcal{H}(t_{i,j})) \Big) \cdot p(@ \geq T \mid \mathcal{H}(t_{I,J_I})) \quad (2)$$

where $p(k_\ell@t_\ell \mid \mathcal{H}(t_\ell))$ is the probability density that the *next* event after $\mathcal{H}(t_\ell)$ occurs at time $t_\ell$ and has type $k_\ell$, and $p(@ \geq T \mid \mathcal{H}(t_\ell))$ is the probability that it occurs at some time $\geq T$.

In this paper, we will attempt to guess $Miss$ by sampling $\mathbf{z}$ values from the posterior distribution

$$p(Miss = \mathbf{z} \mid Obs = \mathbf{x}) \propto p(Comp = \mathbf{x} \sqcup \mathbf{z}) \cdot p_{\text{miss}}(Miss = \mathbf{z} \mid Comp = \mathbf{x} \sqcup \mathbf{z}) \quad (3)$$

---

[1]Our conventions of mathematical notation mainly follow those given by Mei & Eisner (2017, section 2).

We may obviously drop the second factor of equation (3) if (for the given $\mathbf{x}$) it is known to be a constant function of $\mathbf{z}$. In this case, the events are said to be **missing at random (MAR)**. Otherwise, they are **missing not at random (MNAR)** Little & Rubin (1987).

It is often intractable to sample *exactly* from $p(\mathbf{z} \mid \mathbf{x})$, because the first factor $p(\mathbf{x} \sqcup \mathbf{z})$ is complicated (e.g. a neural net). The difficulty is that $\mathbf{x}$ and $\mathbf{z}$ can be interleaved with each other. As an alternative, we can use normalized importance sampling, drawing many $\mathbf{z}$ values from a **proposal distribution** $q(\mathbf{z} \mid \mathbf{x})$ and weighting them in proportion to $\frac{p(\mathbf{z}|\mathbf{x})}{q(\mathbf{z}|\mathbf{x})}$. To make it easy to sample from $q(\mathbf{z} \mid \mathbf{x})$, we adopt the following factored definition:

$$q(\mathbf{z} \mid \mathbf{x}) = \prod_{i=0}^{I} \left( \left( \prod_{j=1}^{J_i} q(k_{i,j} @ t_{i,j} \mid \mathcal{H}(t_{i,j}), \mathcal{F}(t_{i,j})) \right) \cdot q(@ \geq t_{i+1} \mid \mathcal{H}(t_{i,J_i}), \mathcal{F}(t_{i,J_i})) \right) \quad (4)$$

This resembles equation (2), but it conditions each proposed unobserved event not only on the history but also on the **future** $\mathcal{F}(t_{i,j}) \overset{\text{def}}{=} \{k_{i+1} @ t_{i+1}, \ldots, k_{I+1} @ t_{I+1}\}$. This future consists of all the *observed* events that happen after $t_{i,j}$, where in general $\mathcal{F}(t) \overset{\text{def}}{=} \{k_i @ t_i : t < t_i\}$ for any $t \in [0, T]$. Note the asymmetry with $\mathcal{H}(t)$, which includes both observed and unobserved events. $q(\mathbf{z} \mid \mathbf{x})$ can be trained to approximate the target distribution $p(\mathbf{z} \mid \mathbf{x})$, by making $q(\cdot \mid \mathcal{H}, \mathcal{F}) \approx p(\cdot \mid \mathcal{H}, \mathcal{F})$.

We can sample $\mathbf{z}$ from $q(\mathbf{z} \mid \mathbf{x})$ in chronological order: for each $0 \leq i \leq I$ in turn, draw a sequence of $J_i$ unobserved events that follow the observed event $k_i @ t_i$. This sequence ends (thereby determining $J_i$) if the next proposed event would have fallen after $t_{i+1}$ and thus is preempted by the observed event $k_{i+1} @ t_{i+1}$.

Specific $p(\mathbf{x} \sqcup \mathbf{z})$ and $q(\mathbf{z} \mid \mathbf{x})$ distributions will be introduced below.

## 2.1 THE NEURAL HAWKES PROCESS

As our generative model of complete event streams $Comp$ in equation (2), we need a multivariate point process model. We choose the **neural Hawkes process** (Mei & Eisner, 2017), which has proven flexible and effective at modeling many real-world event streams.

Given the history $\mathcal{H}(t)$ of all events before time $t$, the process defines an **intensity** $\lambda_k(t \mid \mathcal{H}(t)) \in \mathbb{R}_{\geq 0}$, which may be thought of as the instantaneous *rate* of events of type $k$ at time $t$. More precisely, as $dt \to 0^+$, the number of events of type $k$ occurring in the interval $[t, t + dt)$, divided by $dt$, approaches $\lambda_k(t \mid \mathcal{H}(t))$. If no event of any type occurs in this interval (which becomes almost sure as $dt \to 0^+$), one may still occur in the next interval $[t + dt, t + 2dt)$, and so on.

The intensity functions $\lambda_k(t \mid \mathcal{H}(t))$ are continuous on intervals during which no event occurs (note that $\mathcal{H}(t)$ is constant on such intervals). They jointly determine a distribution over the time of the next event after $\mathcal{H}(t)$, as used in every factor of equation (2). As it turns out (Mei & Eisner, 2017),

$$\log p(Comp = \mathbf{x} \sqcup \mathbf{z}) = \sum_{\ell:t_\ell < T} \log \lambda_{k_\ell}(t_\ell \mid \mathcal{H}(t_\ell)) - \int_{t=0}^{T} \sum_{k=1}^{K} \lambda_k(t \mid \mathcal{H}(t)) dt \quad (5)$$

We can therefore train the parameters $\boldsymbol{\theta}$ of the $\lambda_k$ functions by maximizing log-likelihood on training data. Each datum is a complete event stream $\mathbf{x} \sqcup \mathbf{z}$ over some time interval $[0, T]$. In practice, we stop training early when log likelihood stops increasing on held-out development data.

The neural Hawkes process specifically parametrizes $\lambda_k(t \mid \mathcal{H}(t))$ as

$$\lambda_k(t \mid \mathcal{H}(t)) = f_k(\mathbf{v}_k^\top \mathbf{h}(t)) \quad (6a)$$

$$f_k(x) = s_k \log(1 + \exp(x/s_k)) \quad (6b)$$

The vector $\mathbf{h}(t) \in (-1, 1)^D$ summarizes $(t, \mathcal{H}(t))$. It is the hidden state of a **continuous-time LSTM** (Mei & Eisner, 2017) that read the events in $\mathcal{H}(t)$ *as they happened*, and then waited until time $t$. The state of such an LSTM evolves endogenously as it waits for the next event, so the *timing* of the past events has been incorporated into the state $\mathbf{h}(t)$.

## 3 PARTICLE METHODS

We now describe our particle-based methods for imputing missing $\mathbf{z}$ (equation (3)). The details are spelled out in Algorithm 1 and Appendix A. For intuition, Figure 1 walks through part of an example.

Algorithm 1 is a **Sequential Monte Carlo (SMC)** approach. It returns an **ensemble of weighted particles** $\mathcal{Z}_M = \{(\mathbf{z}_m, w_m)\}_{m=1}^M$. Each particle $\mathbf{z}_m$ is sampled from the **proposal distribution** $q(\mathbf{z} \mid \mathbf{x})$, which is defined to support sampling via a *sequential* procedure that draws one unobserved event at a time. The corresponding $w_m$ are importance weights, which are defined as follows, but which are also built up one factor at a time within Algorithm 1:

$$w_m = \frac{\tilde{w}_m}{\sum_{m=1}^M \tilde{w}_m} \qquad \text{(7a)} \qquad\qquad \tilde{w}_m = \frac{p(Obs = \mathbf{x}, Miss = \mathbf{z}_m)}{q(\mathbf{z}_m \mid \mathbf{x})} \qquad \text{(7b)}$$

The numerator of (7b) is given by equations (1)–(2).[2] Equation (3) implies that if we could set $q(\mathbf{z} \mid \mathbf{x})$ equal to $p(\mathbf{z} \mid \mathbf{x})$, so that the particles were IID samples from the desired posterior distribution, then all of the $\tilde{w}_m$ would be equal and thus $w_m = 1/m$ for all $m$. In practice, $q$ will not equal $p$, but will be easier than $p$ to sample from. To correct for the mismatch, the importance weights $w_m$ are higher for particles that $q$ proposes more rarely than $p$ would.

The distribution formed by the ensemble, $\hat{p}(\mathbf{z})$, approaches $p(\mathbf{z} \mid \mathbf{x})$ as $M \to \infty$ (Doucet & Johansen, 2009). Thus, for large $M$, the ensemble may be used to estimate the expectation of *any* function $f(\mathbf{z})$, via

$$\mathbb{E}_{p(\mathbf{z}|\mathbf{x})}[f(\mathbf{z})] \approx \mathbb{E}_{\hat{p}}[f(\mathbf{z})] = \sum_{\mathbf{z}} \hat{p}(\mathbf{z})f(\mathbf{z}_m) = \sum_{m=1}^M w_m f(\mathbf{z}_m) \qquad \text{(8)}$$

In the subsections below, we will describe two specific proposal distributions $q$ that are appropriate for the neural Hawkes process. These distributions define intensity functions $\lambda^q$ over time intervals.

The trickiest part of Algorithm 1 (at line 32) is to sample the next unobserved event from the proposal distribution $q$. Here we use the **thinning algorithm** (Lewis & Shedler, 1979; Liniger, 2009; Mei & Eisner, 2017). Briefly, this is a rejection sampling algorithm whose own proposal distribution uses a *constant* intensity $\lambda^*$, making it a homogeneous Poisson process (which is easy to sample from). A event proposed by the Poisson process at time $t$ is accepted with probability $\lambda^q(t)/\lambda^* \leq 1$. If it is rejected, we move on to the next event proposed by the Poisson process, continuing until we either accept such an unobserved event or are preempted by the arrival of the next observed event.

### 3.1 Neural Hawkes Particle Filtering

We already have a neural Hawkes process $p$ that was trained on complete data. This neurally defines an intensity function $\lambda_k^p(t \mid \mathcal{H}(t))$ for *any* history $\mathcal{H}(t)$ of events before $t$ and each event type $k$.

The simplest proposal distribution uses precisely this process to draw the unobserved events. More precisely, for each $i = 0, 1, \ldots, I$, for each $j = 0, 1, 2, \ldots$, we let the next event $k_{i,j+1}@t_{i,j+1}$ be the first event generated by the competing intensity functions $\lambda_k(t \mid \mathcal{H}(t))$ over the interval $t \in (t_{i,j}, t_{i+1})$, where $\mathcal{H}(t)$ consists of all observed and unobserved events up through index $\langle i, j \rangle$. If no event is generated on this interval, then the next event is $k_{i+1}@t_{j+1}$. This method is implemented by Algorithm 1 with $smooth = \textbf{false}$.

### 3.2 Neural Hawkes Particle Smoothing

The neural Hawkes process $p(\mathbf{x} \sqcup \mathbf{z})$ only offers us $\lambda_k(t \mid \mathcal{H}(t))$. Extra machinery is needed to condition on $\mathcal{F}(t)$ as well.

We use a **right-to-left continuous-time LSTM**, whose details will be shown shortly in Appendix B, to summarize the future $\mathcal{F}(t)$ for any time $t$ into another hidden state vector $\bar{\mathbf{h}}(t) \in \mathbb{R}^{D'}$. Then we parameterize the proposal intensity using an extended variant of equation (6a):

$$\lambda_k^q(t \mid \mathcal{H}(t), \mathcal{F}(t)) = f_k(\mathbf{v}_k^\top(\mathbf{h}(t) + \mathbf{V}\bar{\mathbf{h}}(t))) \qquad \text{(9)}$$

---

[2]As discussed earlier, if we are willing to make a MAR assumption, we may omit the $p_{\text{miss}}$ factor of equation (1) because it is constant (though unknown).

This extra machinery is used by Algorithm 1 when $smooth = $ **true**. Intuitively, the left-to-right $\mathbf{h}(t)$, as explained in Mei & Eisner (2017), is supposed to learn sufficient statistics for predicting the future as it reads the complete history $\mathcal{H}(t)$, while the right-to-left $\bar{\mathbf{h}}(t)$ carries back observed information from the future $\mathcal{F}(t)$ in order to correct any mistaken posterior beliefs that $\mathbf{h}(t)$ carries.

The right-to-left LSTM has the same architecture as the left-to-right LSTM used in the neural Hawkes process (Mei & Eisner, 2017), but a separate parameter vector. For any time $t \in (0, T)$, it arrives at $\bar{\mathbf{h}}(t)$ by reading *only* the *observed* events $\{k_i@t_i : t < t_i \leq t_I\}$, i.e., $\mathcal{F}(t)$, in *reverse* chronological order. Formulas are given in Appendix B.

This is very similar to the forward-backward algorithm (Rabiner, 1989) or Kalman smoothing (Rauch et al., 1965). However, it is a trained approximation rather than an exact method because of the complicated neural models involved; see Lin & Eisner (2018) for careful discussion of the connection. Regardless of the chosen model, particle smoothing is to particle filtering as Kalman smoothing is to Kalman filtering (Kalman, 1960; Kalman & Bucy, 1961).

### 3.2.1 LEARNING TO PROPOSE

Training the proposal distribution $q(\mathbf{z} \mid \mathbf{x})$ means learning its parameters $\phi$, namely the parameters of the right-to-left LSTM together with matrix $\mathbf{V}$. We are interested in minimizing the **Kullback-Leibler (KL) divergence** between $q(\mathbf{z} \mid \mathbf{x})$ and $p(\mathbf{z} \mid \mathbf{x})$. Although $p(\mathbf{z} \mid \mathbf{x})$ is unknown, the gradient of **inclusive KL divergence** between $q(\mathbf{z} \mid \mathbf{x})$ and $p(\mathbf{z} \mid \mathbf{x})$ is

$$\nabla_\phi \mathrm{KL}(p\|q) = \mathbb{E}_{\mathbf{z} \sim p(\mathbf{z}|\mathbf{x})}[-\nabla_\phi \log q(\mathbf{z} \mid \mathbf{x})] \tag{10}$$

where $\log q(\mathbf{z} \mid \mathbf{x})$ is analogous to equation (5).The gradient of **exclusive KL divergence** is:

$$\nabla_\phi \mathrm{KL}(q\|p) = \mathbb{E}_{\mathbf{z} \sim q(\mathbf{z}|\mathbf{x})}[\nabla_\phi \left( \frac{1}{2} \left( \log q(\mathbf{z} \mid \mathbf{x}) - \log p(\mathbf{x} \sqcup \mathbf{z}) - p_{\mathrm{miss}}(\mathbf{z} \mid \mathbf{x} \sqcup \mathbf{z}) \right)^2 \right)] \tag{11}$$

where $p(\mathbf{x} \sqcup \mathbf{z})$ is given in equation (5) and $p_{\mathrm{miss}}(\mathbf{z} \mid \mathbf{x} \sqcup \mathbf{z})$ is assumed known to us for any given pair of $\mathbf{x}$ and $\mathbf{z}$.

Minimizing inclusive KL divergence aims at high recall—$q(\mathbf{z} \mid \mathbf{x})$ is adjusted to assign high probabilities to all of the good hypotheses (according to $p(\mathbf{z} \mid \mathbf{x})$). Conversely, minimizing exclusive KL divergence aims at high precision—$q(\mathbf{z} \mid \mathbf{x})$ is adjusted to assign low probabilities to poor reconstructions, so that they will not be proposed. We seek to minimize the linearly combined divergence

$$\mathrm{Div}(p\|q) = \beta \mathrm{KL}(p\|q) + (1 - \beta)\mathrm{KL}(q\|p) \text{ with } \beta \in [0, 1] \tag{12}$$

and our (Adam) training is early-stopped when the divergence stops decreasing on the held-out development set. When tuning our system (Appendix E.2), $\beta = 1$ gave the best results (perhaps unsurprisingly).

But how do we measure these divergences between $q(\mathbf{z} \mid \mathbf{x})$ and $p(\mathbf{z} \mid \mathbf{x})$? Of course, we actually want the *expected* divergence when the observed sequence $\mathbf{x} \sim p$. We pretend that the data are distributed identically to the model $p$, and thus we sample $\mathbf{x}$ from our training examples. For the exclusive divergence, we sample $\mathbf{z} \sim q(\cdot \mid \mathbf{x})$ from our proposal distribution; notice that optimizing $q$ here is essentially the REINFORCE algorithm Williams (1992). For the inclusive divergence, we again pretend that the data are distributed identically to $p$, so instead of sampling a value of $\mathbf{z} \sim p(\cdot \mid \mathbf{x})$, we can simply use the ground-truth $\mathbf{z}$ that happened to occur with this $\mathbf{x}$. We obtain each example with its ground truth $\mathbf{z}$ by starting with a *fully observed* sequence $Comp$ and sampling a partition into $Obs = \mathbf{x}$, $Miss = \mathbf{z}$ from the known missingness mechanism $p_{\mathrm{miss}}$.

Appendix F discusses situations where training on incomplete data by EM is possible.

## 4 A LOSS FUNCTION AND DECODING METHOD

It is sometimes useful to find a *single* hypothesis $\hat{\mathbf{z}}$ that minimizes the *Bayes risk*, i.e., the expected distance from the *unknown* ground truth $\mathbf{z}^*$. This procedure is called **minimum Bayes risk (MBR) decoding** and can be approximated with our ensemble of weighted particles:

$$\hat{\mathbf{z}} = \arg \min_{\mathbf{z} \in \mathcal{Z}} \sum_{\mathbf{z}^* \in \mathcal{Z}} p(\mathbf{z}^* \mid \mathbf{x}) D(\mathbf{z}, \mathbf{z}^*) \approx \arg \min_{\mathbf{z} \in \mathcal{Z}} \sum_{m=1}^{M} w_m D(\mathbf{z}, \mathbf{z}_m) \tag{13}$$

where $D(\mathbf{z}^*, \mathbf{z})$ is the **distance** between $\mathbf{z}^*$ and $\mathbf{z}$. We now propose a specific distance function $D$.

### 4.1 OPTIMAL TRANSPORT DISTANCE

The distance between $\mathbf{z}$ and $\mathbf{z}^*$ is defined as the minimum cost to edit $\mathbf{z}$ into $\mathbf{z}^*$. To accomplish this edit, we must identify the best **alignment**—a one-to-one partial matching $\mathbf{a}$—of the events in the two sequences. We require any two aligned events to have the same type $k$. An alignment edge between a predicted event at time $t$ (in $\mathbf{z}$) and a true event at time $t^*$ (in $\mathbf{z}^*$) incurs a cost of $|t - t^*|$ to move the former to the correct time. Each unaligned event in $\mathbf{z}$ incurs a deletion cost of $C_{\text{delete}}$, and each unaligned event in $\mathbf{z}^*$ incurs an insertion cost of $C_{\text{insert}}$. Thus,

$$D(\mathbf{z}^*, \mathbf{z}) = \min_{\mathbf{a} \in \mathcal{A}(\mathbf{z}^*, \mathbf{z})} D(\mathbf{z}^*, \mathbf{z}, \mathbf{a}) \tag{14}$$

where $\mathcal{A}(\mathbf{z}^*, \mathbf{z})$ is the set of all possible alignments between $\mathbf{z}^*$ and $\mathbf{z}$. Finding this distance (and its corresponding alignment $\mathbf{a}$) is similar to finding the edit distance or dynamic time warping between two discrete-time sequences, and a similar dynamic programming algorithm is presented in Algorithm 2 of Appendix C. We set insertion and deletion costs to be the same, i.e., $C_{\text{insert}} = C_{\text{delete}} = C$, so that the distance is symmetric.

### 4.2 APPROXIMATE MBR DECODING

Since aligned events must have the same type, the MBR problem decomposes into *separately* choosing a set $\hat{\mathbf{z}}^{(k)}$ of type-$k$ events for each $k = 1, 2, \ldots, K$, based on the particles' sets $\mathbf{z}_m^{(k)}$ of type-$k$ events. Thus, we simplify the presentation by omitting $^{(k)}$ throughout this section.

Finding the optimal set $\hat{\mathbf{z}}$ appears to be NP-hard, by analogy with the Steiner string problem. It involves searching over the infinitely many $\mathbf{z} \in \mathcal{Z}$. Fortunately, the optimal transport distance $D$ define in section 4.1 warrants:

**Theorem 1.** *Given $\{\mathbf{z}_m\}_{m=1}^M$, if we define $\mathbf{z}_\sqcup = \bigsqcup_{m=1}^M \mathbf{z}_m$, then:*

$$\exists \hat{\mathbf{z}} \in \mathcal{P}(\mathbf{z}_\sqcup) \text{ such that } \sum_{m=1}^M w_m D(\mathbf{z}_m, \hat{\mathbf{z}}) = \min_{\mathbf{z} \in \mathcal{Z}} \sum_{m=1}^M w_m D(\mathbf{z}_m, \mathbf{z}) \tag{15}$$

*where $\mathcal{P}(\mathbf{z})$ is the power set of any given $\mathbf{z}$—the set of subsequences of $\mathbf{z}$ (including empty sequence and $\mathbf{z}$ itself). That is to say, there exists one subsequence of $\mathbf{z}_\sqcup$ that achieves the minimum Bayes risk (or the minimum weighted distance).*

Why this is true? Let's suppose $t$ is the time of one event in $\mathbf{z}^*$, and it is aligned to one event at $t_1$ in $\mathbf{z}_\sqcup$. Recall that the alignment cost between these two events is $|t - t_1|$, so we can simply change $\mathbf{z}^*$ by moving $t$ to $t_1$ to minimize its total alignment cost. Then what if it is aligned to events at $t_1 < t_2$? While $t < t_1$, increasing $t$ always decreases its total alignment cost $|t - t_1| + |t - t_2| = t_1 + t_2 - 2t$. Similarly, we should decrease $t$ if $t > t_2$. When $t_1 \le t \le t_2$, the alignment cost is fixed at $t_2 - t_1$. Therefore, the total alignment cost of this event is always minimized if $t$ is moved to $t_1$ or $t_2$. This argument easily generalizes to the cases where $t$ is aligned to more aligned events. The generalization to the cases that each alignment has a weight $w$ is also straightforward, and the full proof can be found in Appendix D.1.

Now we have simplified this decoding problem as a combinational optimization problem:

$$\hat{\mathbf{z}} = \arg\min_{\mathbf{z} \in \mathcal{P}(\mathbf{z}_\sqcup)} \sum_{m=1}^M w_m D(\mathbf{z}_m, \mathbf{z}) \tag{16}$$

which can be *approximately* solved although it is still NP-hard to *exactly* solve.

Our algorithm (details in Algorithm 3 of Appendix D) finds $\hat{\mathbf{z}}$ by *iteratively* (1) finding its optimal alignment $\mathbf{a}_m$ with each $\mathbf{z}_m$ (called **Align Phase**) using this method of section 4.1, and then (2) going through the following phases—each of them will update $\hat{\mathbf{z}}$ and decreases the weighted distance $\sum_{m=1}^M w_m D(\mathbf{z}_m, \hat{\mathbf{z}}, \mathbf{a}_m)$ which is an upper bound of $\sum_{m=1}^M w_m D(\mathbf{z}_m, \hat{\mathbf{z}})$:

**Move Phase** Thanks to Theorem 1, given fixed $|\hat{\mathbf{z}}|$ and $\{\mathbf{a}_m\}_{m=1}^M$, $\sum_{m=1}^M w_m D(\mathbf{z}_m, \hat{\mathbf{z}}, \mathbf{a}_m)$ can be minimized by simply moving each $t_i$ in the current $\hat{\mathbf{z}}$ to one of the times, if there is any, which it is aligned to. Otherwise, we keep $t_i$ unchanged.

**Delete Phase** Then we may delete any event in $\hat{\mathbf{z}}$ and its associated element in each $\mathbf{a}_m$ if $\sum_{m=1}^{M} w_m D(\mathbf{z}_m, \hat{\mathbf{z}}, \mathbf{a}_m)$ is decreased afterwards. Note that deletion of each event does not depend on one another, so the weighted distance can actually be minimized by independently checking each event in $\hat{\mathbf{z}}$. The phase tends to discard the events that are aligned to far-apart times (or nowhere).

**Insert Phase** Then we may further decrease $\sum_{m=1}^{M} w_m D(\mathbf{z}_m, \hat{\mathbf{z}}, \mathbf{a}_m)$ by inserting to $\hat{\mathbf{z}}$ some events that can be aligned, at low cost, to those in $\mathbf{z}_m$ which would have been left alone otherwise. Thanks again to Theorem 1, such fortunate events can be chosen from $\bigsqcup_{m=1}^{M} \mathbf{z}_m$.

Note that even though we used $D(\mathbf{z}_m, \hat{\mathbf{z}}, \mathbf{a}_m)$ multiple times in the above description, only the one time in Move Phase needs to actually call the dynamic programming algorithm (section 4.1) to get $\mathbf{a}_m$. Details of the full algorithm (and its theoretical guarantee) can be found in Appendix D.

## 5 EXPERIMENTS

We compare our particle smoothing method with the particle filtering baseline on multiple real-world and synthetic datasets. See Appendix E for training details (e.g., hyperparameter selection).

### 5.1 DATASETS

The datasets that we use in this paper range from short sequences with mean length 15 to long ones with mean length $> 300$. For each of the datasets, we have fully observed data that can be used for training. For each dev and test example, we held out some events from the fully observed sequence, so we present the $\mathbf{x}$ part as input to the proposal distribution but we also know the $\mathbf{z}$ part for evaluation purposes. The dataset and preparation details can be found in Appendix E.

**Synthetic Datasets** We first checked that we could successfully impute unobserved events that are generated from *known* distributions. That is, when the generating distribution actually is a neural Hawkes process, could our method outperform the particle filtering in practice? Is the performance consistent over multiple datasets drawn from different processes? To investigate this, we synthesized 10 datasets, each of which was drawn from a different neural Hawkes process with randomly sampled parameters.

**Elevator System Dataset** (Crites & Barto, 1996). A multi-floor building is often equipped with multiple elevator cars that follow *cooperative* strategies to transport passengers between floors (Lewis, 1991; Bao et al., 1994; Crites & Barto, 1996). Observing the activities of some of the cars might help us impute those of the others. This domain is particularly interesting because it is representative of many real-world cooperative (or competitive) scenarios.

**New York City Taxi Dataset** (Whong, 2014). Each medallion taxi of New York City forms a sequence of time-stamped pick-up and drop-off events in the five boroughs (i.e., Manhattan, Brooklyn, Queens, The Bronx, and Staten Island). Having observed a sequence of drop-off events, it is interesting to see if the proposed method is able to impute the pick-up events (Figure 1).

### 5.2 EVALUATION AND RESULTS

First, as an internal check, we measure *how probable* each ground truth reference $\mathbf{z}^*$ is under the proposal distribution constructed by each method, i.e., $\log q(\mathbf{z}^* \mid \mathbf{x})$. The results on the 12 datasets are displayed in Figure 2.

We now make predictions by MBR decoding. Figure 3 plots the improved performance of neural Hawkes particle smoothing (red) vs. particle filtering (blue).[3] It shows the optimal transport distance broken down by (a) how well it is doing at predicting *which* events happen—measured by the total number of insertions and deletions (x-axis); and (b) how well it is doing at predicting *when* those events happen—measured by the total move cost (y-axis). Different choices of $C$ yield different trade-offs between these two metrics. Intuitively, when $C \approx 0$, the decoder is free to insert and

---

[3]We show the 2 real datasets only. The figures for the 10 synthetic datasets are boringly similar to these.

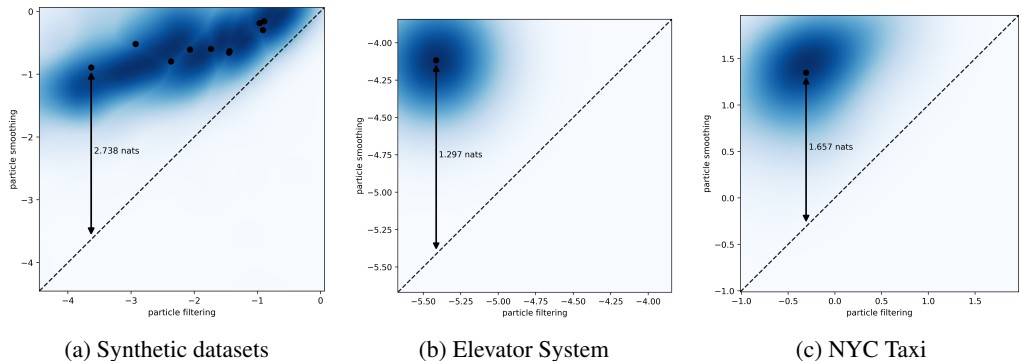

(a) Synthetic datasets      (b) Elevator System      (c) NYC Taxi

Figure 2: Scatterplots of neural Hawkes particle smoothing (y-axis) vs. particle filtering (x-axis). Each point represents a single test sequence, and compares the values of $\log q(\mathbf{z}^* \mid \mathbf{x})/|\mathbf{z}^*|$ (i.e., nats per unobserved event) under the two proposal distributions. Larger numbers mean that the proposal distribution is better at proposing the ground truth. Each dataset's scatterplot is converted to a cloud using kernel density estimation, with the centroid denoted by a black dot. A double-arrowed line indicates the improvement of particle smoothing over filtering. For the synthetic datasets, we draw ten clouds on the same figure and show the line for the set where smoothing improves the most. Particle smoothing performs well even on the datasets where particle filtering performs badly. As we can see, the density function is always well concentrated above $y = x$. That is, this is not merely an average improvement: nearly *every* ground truth sequence increases in proposal probability!

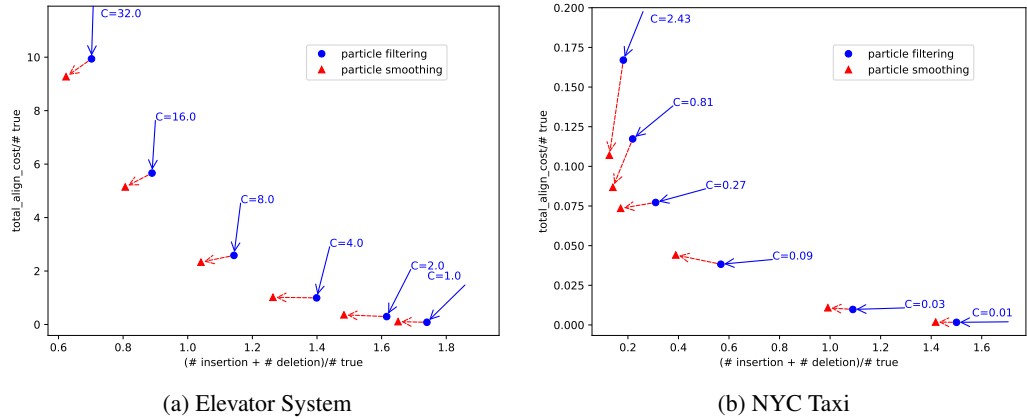

(a) Elevator System      (b) NYC Taxi

Figure 3: Optimal transport distance of neural Hawkes particle smoothing (▲) vs. particle filtering (●) on test data, broken down by the number of insertions and deletions (x-axis) and the total move/alignment cost (y-axis). Both axes are normalized by the number of true events in the test dataset, $\sum_{n=1}^{N} |\mathbf{z}^*{}_n|$. On each dataset, for each $C$, the ● achieves an optimal transport distance that is broken down to the x-axis and y-axis. The ⟶ pointing to ● indicates the gradient direction along which the $D$ function would like us to improve for this $C$ value. The - -> starting from this ● shows the actual improvement obtained by switching to particle smoothing (which is, indeed, an improvement because it has positive dot product with ⟶). Overall, the Pareto frontier (convex hull) of the ▲ symbols dominates the Pareto frontier of the ● symbols, lying everywhere to its left.

delete event tokens, so none of them is moved. As the cost $C$ increases, fewer event tokens are inserted or deleted (so lower cost on the x-axis), but more event tokens are aligned (so higher cost on the y-axis).

# 6 DISCUSSION

Our technical contribution is threefold. First of all, as far as we know, we are the first to develop general sequential Monte Carlo methods (Moral, 1997; Liu & Chen, 1998; Doucet et al., 2000; Doucet & Johansen, 2009) to approximate a target posterior distribution $q(\mathbf{z} \mid \mathbf{x})$ that is based on a neural model of the complete sequence $\mathbf{x} \sqcup \mathbf{z}$ (such as the neural point processes of Du et al. (2016) and Mei & Eisner (2017)). Most similar to our work is Linderman et al. (2017)'s sequen-

tial Monte Carlo method. They modeled complete sequences by a Hawkes process with latent variables—substituting our neural Hawkes process would obtain exactly our particle filtering method (section 3.1). However, our full method has a particle smoother that takes future observations into account while proposing events at any time $t$, which is not considered in Linderman et al. (2017). Shelton et al. (2018) developed a reversible jump Markov chain Monte Carlo (MCMC) sampler (Green & Hastie, 2009) for a Hawkes process with latent variables and it allows future observations to "reach back and suggest possible events earlier in the timeline". But their method takes advantage of the Poisson cluster process representation of Hawkes process—each past event generates a Poisson process with exponentially decaying intensity and each new event belongs to one of the processes. Such a representation cannot be established for a neural point process under which the sequence of all past events as a whole generates only one Poisson process, so their method cannot adapt to the complicated neural model that we consider in this work. Other work that infers unobserved events in continuous time also assumes that complete sequences follow a model that is easier to handle than a neural model, including those based on Markov jump processes (Rao & Teh, 2012; 2013) and continuous-time Bayesian networks (Fan et al., 2010). Lin & Eisner (2018) design a neural particle smoothing algorithm that performs dynamic-programming-style approximate inference on a neural sequential model. But they only work on discrete-time sequences; our neural Hawkes particle smoothing can be seen as a continuous-time generalization of their method.

Secondly, we define the optimal transport distance between event sequences, which is a provably valid metric. It is a Wasserstein distance (Villani, 2008), or Earth Mover's distance (Kantorovitch, 1958; Levina & Bickel, 2001), generalized to unnormalized "distributions". There is more than one way to make this generalization, and this is still a subject of active research (Benamou, 2003; Chizat et al., 2015; Frogner et al., 2015; Chizat et al., 2018). Our definition allows event insertion and deletion while aligning them, but these operations can only apply to an entire event—we cannot align half of an event and delete the other half. Due to these constraints, a dynamic programming rather than linear programming (relaxation) is needed to find the optimal transport. Xiao et al. (2017) also proposed an optimal transport distance between event sequences and it also allows event insertion and deletion. However, their insertion and deletion cost depends on where it is on the time axis while ours doesn't.[4][5] Our optimal transport distance is similar to the **dynamic time warping (DTW)** and its extensions (Sakoe & Chiba, 1971; Listgarten et al., 2005),which is also a method to calculate an optimal match between sequences. But major difference indeed exists: 1) DTW aligns each event to at least one event and does not allow insertion or deletion, while ours aligns each to at most one and allows insertion and deletion; 2) DTW forces the first-to-first and last-to-last alignment while ours does not; 3) DTW does not allow crossing edges while our cost function is such that within the matching for type $k$, there is always an optimal solution with no crossing edges (although another equally good solution may have some).

Last but not least, we design an algorithm to find a sequence of unobserved events whose expected distance to an unknown ground truth reference is approximately minimized, and this reference follows a distribution approximated by a set of (weighted) particles. This problem is similar to finding a consensus representation (Steiner sequence) of a set of sequences, which is described in Gusfield (1997) through the concept of multiple sequence alignment (MSA) (Mount, 2004) in computational biology. The latter is usually solved by progressive alignment construction using a guide tree (Feng & Doolittle, 1987; Larkin et al., 2007; Notredame et al., 2000) and iterative realignment of the initial sequences with addition of new sequences to the growing MSA (Hirosawa et al., 1995; Gotoh, 1996). However, these methods cannot be directly applied to our setting because each event in our sequence is a $k_i@t_i$ pair, not just a discrete type $k_i$—when two events are aligned, we would like their times as well as their types to match.

On multiple synthetic and real-world datasets, our method turns out to be effective at inferring the ground-truth sequence of unobserved events. The improvement of particle smoothing upon particle filtering is substantial and consistent, showing the benefit of training a proposal distribution.

---

[4]This dependence actually makes the distance unintuitive. For example, on interval $[0, 100]$, the distance between $\mathbf{z}_1 = 1, 3, 4, 5$ and $\mathbf{z}_2 = 3, 4, 5$ is $|1 - 3| + |3 - 4| + |4 - 5| + |5 - 100| = 99$ under their definition while it can be small under our definition (e.g. 0.5 if our insertion and deletion cost $C = 0.5$). The latter seems more natural because these two sequences are almost identical.

[5]Besides optimal transport, Stein's method (Stein et al., 1972) is also used to find (or bound) the distance of point processes (Schuhmacher & Xia, 2008; Decreusefond et al., 2016).

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

# Appendices

## A   SEQUENTIAL MONTE CARLO DETAILS

Our main algorithm is presented as Algorithm 1. It covers both particle filtering and particle smoothing, with optional multinomial resampling.

In this section, we also provide some further notes that are not covered in the pseudocode.

**Managing LSTM state information**   In Algorithm 1, when we push events to stacks $\mathcal{H}_m$ and $\mathcal{F}$, we update the respective LSTM's configurations (including gates, cell memories and states), and we revert these updates when we pop events from $\mathcal{F}$. These operations facilitate the computation of intensities $\lambda_k^p(t)$ and $\lambda_k^q(t)$.

**Integral computation**   Computing the weight $w_m$ requires handling the integral on line 28 and line 40 of Algorithm 1. We use the same trick in section B.2 of Mei & Eisner (2017) that gives an unbiased estimate of the integrals by evaluating $(t_{i,j} - t_{i,j-1})\lambda^p(t)$ and $(t_{i,j} - t_{i,j-1})\lambda^q(t)$ at a random $t \sim \mathrm{Unif}(t_{i,j-1}, t_{i,j})$. We can draw $N$ samples instead of one, and this Monte Carlo algorithm will average over these samples to reduce the variance of this noisy estimator.

**Choice of $\lambda^*$**   How do we construct the upper bound $\lambda^*$ (line 33 of Algorithm 1)? For particle filtering, we follow the recipe in B.3 of Mei & Eisner (2017): we can express $\lambda^* = f_k(\max_t g_1(t) + \ldots + \max_t g_n(t))$ where each summand $v_{kd}h_d(t) = v_{kd} \cdot o_{id} \cdot (2\sigma(2c_d(t)) - 1)$ is upper-bounded by $\max_{c \in \{c_{id}, \bar{c}_{id}\}} v_{kd} \cdot o_{id} \cdot (2\sigma(2c) - 1)$. Note that the coefficients $w_{kd}$ may be either positive or negative.

For particle smoothing, we simply have more summands inside $f_k$ so $\lambda^* = f_k(\max_t g_1(t) + \ldots + \max_t g_n(t) + \max_t \bar{g}_1(t) + \ldots + \max_t \bar{g}_{\bar{n}}(t))$ where each extra summand $u_{kd}\bar{h}_d(t) = u_{kd} \cdot o_{id} \cdot (2\sigma(2c_d(t)) - 1)$ is upper-bounded by $\max_{c \in \{c_{id}, \bar{c}_{id}\}} u_{kd} \cdot o_{id} \cdot (2\sigma(2c) - 1)$ and each $u_{kd}$ is the $d$-th element of vector $\mathbf{v}_k^\top \mathbf{V}$ (equation (9)). Note that the $o_{id}, c_{id}, \bar{c}_{id}$ of newly added summands $\bar{g}$ are actually from the right-to-left LSTM while those of $g$ are from the left-to-right LSTM. We only use the same notation here for presentation simplicity.

**Missing data factors in $p$**   Recall that the joint model (1) includes a factor $p_{\mathrm{miss}}(Miss = \mathbf{z} \mid Comp = \mathbf{x} \sqcup \mathbf{z})$, which appears in the numerator of the unnormalized importance weight (7b). Regardless of the form of this factor, it could be multiplied into the particle's weight $\tilde{w}_m$ at the end of sampling (line 15). However, Algorithm 1 assumes a missingness mechanism where the missingness of each event $k@t$ depends only on that event and preceding events,[6] so that $p_{\mathrm{miss}}(Miss = \mathbf{z} \mid Comp = \mathbf{x} \sqcup \mathbf{z})$ factors as

$$\prod_{\ell \in \mathrm{indices}(\mathbf{x} \sqcup \mathbf{z})} p_{\mathrm{miss}}((k_\ell@t_\ell \in Miss) = (k_\ell@t_\ell \in \mathbf{z}) \mid \{k_{\ell'}@t_{\ell'} : \ell' \leq \ell\}) \tag{17}$$

Algorithm 1 can thus *incrementally* incorporate the subfactors of equation (17), and does so at line 30.

**Optional missing data factors in $q$**   We can optionally improve the particle filtering proposal intensities to incorporate the $p_{\mathrm{miss}}$ factor discussed above (in which case that factor will be multiplied into the denominator of (7b) and not just the numerator). This makes $q(\mathbf{z} \mid \mathbf{x})$ better match $p(\mathbf{z} \mid \mathbf{x})$: it means we will rarely posit an unobserved event that would rarely have gone missing.

Specifically, if a completed-data event $k@t$ would have probability $r_k(t \mid \mathcal{H}(t))$ of going missing given the preceding events $\mathcal{H}(t)$, it is wise to define $\lambda_k^q(t \mid \mathcal{H}(t)) = \lambda_k^p(t \mid \mathcal{H}(t)) \cdot r_k(t \mid \mathcal{H}(t))$.

We include this extra $r_k$ factor in our experiments (section 5), although it is not shown in Algorithm 1. It is particularly simple in our experiments, where $r_k$ is constant at 1 or 0 depending on

---

[6]This assumption could trivially be relaxed to allow it to also depend on the missingness of the preceding events, and/or on the future observed events $\mathcal{F}(t)$.

$k$. In other words, some event types $k$ are deterministically missing, while others are *never missing and thus we never propose them as part of* $\mathbf{z}$.

The above considers the particle filtering case. In the case of particle smoothing, we have already tried to ensure by other means that the proposal distribution will incorporate $p_{\text{miss}}$. That is because section 3.2.1 aims to train $\lambda_k^q(t \mid \mathcal{H}(t), \mathcal{F}(t))$ so that the resulting $q(\mathbf{z} \mid \mathbf{x}) \approx p(\mathbf{z} \mid \mathbf{x})$, and the posterior distribution $p(\mathbf{z} \mid \mathbf{x})$ does condition on the missingness of $\mathbf{z}$. However, if the $r_k$ factor is known, why not include it explicitly instead of having to train the BiLSTM to mimic it? Thus it can be convenient to modify the right-hand side of equation (9) to include a factor of $r_k$. This yields a more expressive and better-factored family of proposal distributions: missingness is now handled by the known $r_k$ factor and the BiLSTM does not have to explain it.

Modifying equation (9) in this way is particularly useful in the special case $r_k = 0$ (i.e., event type $k$ is never missing and should not be proposed). There, it enforces the hard constraint that $\lambda_k^q = 0$ (something that the BiLSTM by itself could not achieve); and since this constraint is enforced regardless of the BiLSTM parameters, the events of type $k$ appropriately become irrelevant to the training of the BiLSTM, which can focus on predicting other event types. We do this in our experiments.

## B  RIGHT-TO-LEFT CONTINUOUS-TIME LSTM

Here we give details of the right-to-left LSTM from section 3.2. At each time $t \in (0, T)$, its hidden state $\bar{\mathbf{h}}(t)$ is continually obtained from the memory cells $\mathbf{c}(t)$ as the cells decay:

$$\bar{\mathbf{h}}(t) = \mathbf{o}_i \odot (2\sigma(2\mathbf{c}(t)) - 1) \text{ for } t \in (t_{i-1}, t_i] \tag{18}$$

where the interval $(t_{i-1}, t_i]$ has consecutive observations $k_{i-1}@t_{i-1}$ and $k_i@t_i$ as endpoints. At $t_i$, the continuous-time LSTM reads $k_i@t_i$ and updates the current (decayed) hidden cells $\mathbf{c}(t)$ to new initial values $\mathbf{c}_{i-1}$, based on the current (decayed) hidden state $\bar{\mathbf{h}}(t_i)$, as follows:[7]

$$\mathbf{i}_{i-1} \leftarrow \sigma\left(\mathbf{W}_i \mathbf{k}_i + \mathbf{U}_i \mathbf{h}(t_i) + \mathbf{d}_i\right) \tag{19a}$$

$$\mathbf{f}_{i-1} \leftarrow \sigma\left(\mathbf{W}_f \mathbf{k}_i + \mathbf{U}_f \mathbf{h}(t_i) + \mathbf{d}_f\right) \tag{19b}$$

$$\mathbf{z}_{i-1} \leftarrow 2\sigma\left(\mathbf{W}_z \mathbf{k}_i + \mathbf{U}_z \mathbf{h}(t_i) + \mathbf{d}_z\right) - 1 \tag{19c}$$

$$\mathbf{o}_{i-1} \leftarrow \sigma\left(\mathbf{W}_o \mathbf{k}_i + \mathbf{U}_o \mathbf{h}(t_i) + \mathbf{d}_o\right) \tag{19d}$$

$$\mathbf{c}_{i-1} \leftarrow \mathbf{f}_{i-1} \odot \mathbf{c}(t_i) + \mathbf{i}_{i-1} \odot \mathbf{z}_{i-1} \tag{20a}$$

$$\bar{\mathbf{c}}_{i-1} \leftarrow \bar{\mathbf{f}}_{i-1} \odot \bar{\mathbf{c}}_i + \bar{\mathbf{i}}_{i-1} \odot \mathbf{z}_{i-1} \tag{20b}$$

$$\boldsymbol{\delta}_{i-1} \leftarrow f\left(\mathbf{W}_d \mathbf{k}_i + \mathbf{U}_d \mathbf{h}(t_i) + \mathbf{d}_d\right) \tag{20c}$$

The vector $\mathbf{k}_i \in \{0, 1\}^K$ is the $i^{\text{th}}$ input: a one-hot encoding of the new event $k_i$, with non-zero value only at the entry indexed by $k_i$. Then, $\mathbf{c}(t)$ is given by (21), which continues to control $\mathbf{h}(t)$ except that $i$ has now decreased by 1).

$$\mathbf{c}(t) \stackrel{\text{def}}{=} \bar{\mathbf{c}}_{i-1} + (\mathbf{c}_{i-1} - \bar{\mathbf{c}}_{i-1}) \exp\left(-\boldsymbol{\delta}_{i-1}(t_i - t)\right) \text{ for } t \in (t_{i-1}, t_i] \tag{21}$$

On the interval $[t_{i-1}, t_i)$, $\mathbf{c}(t)$ follows an exponential curve that begins at $\mathbf{c}_{i-1}$ (in the sense that $\lim_{t \to t_i^-} \mathbf{c}(t) = \mathbf{c}_{i-1}$) and decays, as time $t$ decreases, toward $\bar{\mathbf{c}}_{i-1}$.

## C  OPTIMAL TRANSPORT DISTANCE DETAILS

In this section, we first present the detailed algorithm of finding the OTD and its corresponding alignment, and then prove it is a valid metric.

### C.1  ALGORITHM DETAILS

The details of how to find optimal transport distance is presented in Algorithm 2.

### C.2  PROOF THAT OPTIMAL TRANSPORTATION DISTANCE IS A VALID METRIC

We have defined that the optimal transportation distance (OTD), and we prove that it is a valid metric here.

---

[7]The upright-font subscripts i, f, z and o are not variables, but constant labels that distinguish different $\mathbf{W}$, $\mathbf{U}$ and $\mathbf{d}$ tensors. The $\bar{\mathbf{f}}$ and $\bar{\mathbf{i}}$ in equation (20b) are defined analogously to $\mathbf{f}$ and $\mathbf{i}$ but with different weights.

---

**Algorithm 1** Sequential Monte Carlo — Neural Hawkes Particle Filtering/Smoothing

---

**Input:** observed seq. $\mathbf{x} = k_0@t_0, \ldots, k_{I+1}@t_{I+1}$ with $k_0 = \textsc{bos}, t_0 = 0, k_{I+1} = \textsc{eos}, t_{I+1} = T$; model $p$; missingness mechanism $p_{\text{miss}}$; proposal distribution $q$; number of particles $M$; boolean flags *smooth* and *resample*

**Output:** collection $\{(\mathbf{z}_1, w_1), \ldots, (\mathbf{z}_M, w_M)\}$ of weighted particles

1: **procedure** SEQUENTIALMONTECARLO($\mathbf{x}, p, p_{\text{miss}}, q, M, smooth, resample$)
2:   **for** $m = 1$ **to** $M$ :   ▷ *init weighted particles $(\mathbf{z}_m, w_m)$. History $\mathcal{H}_m$ combines $\mathbf{z}_m$ with a prefix of $\mathbf{x}$*
3:     $\mathbf{z}_m \leftarrow$ empty sequence; $w_m \leftarrow 1$; $\mathcal{H}_m \leftarrow$ empty stack
4:   **if** *smooth* :                                                    ▷ *use particle smoothing*
5:     $\mathcal{F} \leftarrow$ empty stack
6:     **for** $i = I$ **downto** $0$ :                  ▷ *stack of all future observed events*
7:       push $k_{i+1}@t_{i+1}$ onto $\mathcal{F}$ ▷ *as we reach these events, we'll pop from $\mathcal{F}$ and push onto $\mathcal{H}_m$ ($\forall m$)*
8:   **else**                                            ▷ *use particle filtering instead*
9:     $\mathcal{F} \leftarrow$ **ignored**     ▷ *special value if we're not using the future stack; unaffected by* pop *operation*
10:   **for** $i = 0$ **to** $I$ :  ▷ *observe present event $k_i@t_i$, then propose unobserved events on interval $(t_i, t_{i+1})$*
11:     **for** $m = 1$ **to** $M$ :
12:       DRAWSEGMENT($i, m$)         ▷ *destructively extend $\mathbf{z}_m, w_m, \mathcal{H}_m$ with events on $[t_i, t_{i+1})$*
13:     pop $\mathcal{F}$                      ▷ *pop $k_{i+1}@t_{i+1}$ from $\mathcal{F}$: it's no longer in the future but in the present*
14:     **if** *resample* : RESAMPLE()   ▷ *optional multinomial resampling replaces all weighted particles*
15:   **return** $\{(\mathbf{z}_m, w_m / \sum_{m=1}^{M} w_m)\}_{m=1}^{M}$   ▷ *M particles with weights normalized as in equation* (7a)
16: **procedure** RESAMPLE                            ▷ *has access to global variables*
17:   **for** $m = 1$ **to** $M$ : ▷ *often draws multiple copies of good (high-weight) particles, 0 copies of bad ones*
18:     $\tilde{\mathbf{z}}_m \sim \text{Categorical}(\{\mathbf{z}_m \mapsto w_m / \sum_{m=1}^{M} w_m\}_{m=1}^{M})$
19:   **for** $m = 1$ **to** $M$ :
20:     $\mathbf{z}_m \leftarrow \tilde{\mathbf{z}}_m$; $w_m \leftarrow 1$                      ▷ *update particles and their weights*
21: **procedure** DRAWSEGMENT($i, m$)                   ▷ *has access to global variables*
22:   ▷ *$p$ gives info to define function $\lambda_k^p(t) \stackrel{\text{def}}{=} \lambda_k(t \mid \mathcal{H}_m)$*
23:   ▷ *$q$ gives info to define function $\lambda_k^q(t) \stackrel{\text{def}}{=} \lambda_k(t \mid \mathcal{H}_m, \mathcal{F})$, or $\lambda_k^q(t) = \lambda_k^p(t)$ if $\mathcal{F} = $ **ignored***
24:   ▷ *these functions consult state of left-to-right LSTM that's read $\mathcal{H}_m$ & right-to-left LSTM that's read $\mathcal{F}$*
25:   ▷ *we also define the **total intensity functions** $\lambda^p(t) \stackrel{\text{def}}{=} \sum_{k=1}^{K} \lambda_k^p(t)$ and $\lambda^q(t) \stackrel{\text{def}}{=} \sum_{k=1}^{K} \lambda_k^q(t)$*
26:   $j \leftarrow 0$; $k_{i,j} \leftarrow k_i$; $t_{i,j} \leftarrow t_i$                     ▷ *ready to observe $i^{th}$ event of $\mathbf{x}$*
27:   **while true** :  ▷ *one iteration adds event at index $\langle i, j \rangle$ (for $j = 0, 1, 2, \ldots$ until we break out of loop)*
28:     $w_m \leftarrow w_m \cdot \lambda_{k_{i,j}}^p(t_{i,j}) \exp\left(-\int_{t'=t_{i,j-1}}^{t_{i,j}} \lambda^p(t')dt'\right)$     ▷ *new factor in numerator $p$ of* (7b)
29:     push $k_{i,j}@t_{i,j}$ onto $\mathcal{H}_m$; $t \leftarrow t_{i,j}$     ▷ *event just generated by $p$ now becomes part of history*
30:     $w_m \leftarrow w_m \cdot p_{\text{miss}}((k_{i,j}@t_{i,j} \in Miss) = (j > 0) \mid \mathcal{H}_m)$   ▷ *new factor in numerator $p$ of* (7b)
31:     ▷ *Now draw possible missing event at index $\langle i, j+1 \rangle$; we'll loop back and add it if it falls in $(t_i, t_{i+1})$*
32:     **repeat**                                ▷ *thinning algorithm (see Mei & Eisner, 2017)*
33:       find any $\lambda^* \geq \sup\{\lambda^q(t') : t' \in (t, t_{i+1})\}$       ▷ *e.g., old $\lambda^*$ still works if $i$ unchanged*
34:       draw $\Delta \sim \text{Exp}(\lambda^*)$, $u \sim \text{Unif}(0, 1)$
35:       $t \mathrel{+}= \Delta$                           ▷ *time of next proposed event (before thinning)*
36:       **if** $t \geq t_{i+1}$ :      ▷ *proposed event falls outside $(t_i, t_{i+1})$, where $k_{i+1}@t_{i+1}$ is top element of $\mathcal{F}$*
37:         **return**               ▷ *done adding events on $[t_i, t_{i+1})$; time to break out of* **while** *loop*
38:     **until** $u\lambda^* \leq \lambda^q(t)$               ▷ *thinning: accept proposal with prob $\frac{\lambda^q(t)}{\lambda^*} \leq 1$*
39:     $j \leftarrow j + 1$; $t_{i,j} \leftarrow t$; $k_{i,j} \sim \text{Categorical}(\{k \mapsto \frac{\lambda_k^q(t)}{\lambda^q(t)}\}_{k=1}^{K})$; append $k_{i,j}@t_{i,j}$ to $\mathbf{z}_m$
40:     $w_m \leftarrow w_m / \left(\lambda_{k_{i,j}}^q(t_{i,j}) \exp(-\int_{t'=t_{i,j-1}}^{t_{i,j}} \lambda^q(t')dt')\right)$   ▷ *new factor in denominator $q$ of* (7b)

---

It's trivial that OTD is non-negative, since movement, deletion and insertion costs are all positive.

It's also trivial to prove that the following statement is true:

$$D(\mathbf{z}_1, \mathbf{z}_2) = 0 \Leftrightarrow \mathbf{z}_1 = \mathbf{z}_2, \tag{22}$$

where $\mathbf{z}_1$ and $\mathbf{z}_2$ are two sequences. If $\mathbf{z}_1$ is not identical to $\mathbf{z}_2$, the distance of them must be larger than 0 since we have to do some movement, insertion or deletion to make them exactly matched, so the right direction of equation (22) holds. If the distance between $\mathbf{z}_1$ and $\mathbf{z}_2$ is zero, which means

---

**Algorithm 2** A Dynamic Programming Algorithm to Find Optimal Transport Distance

---

**Input:** proposal $\hat{\mathbf{z}}$; reference $\mathbf{z}^*$
**Output:** optimal transport distance $d$; alignment $\mathbf{a}$
1: **procedure** OTD($\mathbf{z}^*, \hat{\mathbf{z}}$)
2:    $d \leftarrow 0; \mathbf{a} \leftarrow$ empty collection $\{\}$
3:    **for** $k \leftarrow 1$ **to** $K$ :
4:      $d^{(k)}, \mathbf{a}^{(k)} \leftarrow$ DYNAMICPROGRAMMING($\hat{\mathbf{z}}^{(k)}, \mathbf{z}^{*(k)}$)
5:      $d \leftarrow d + d^{(k)}; \mathbf{a} \leftarrow \mathbf{a} \cup \mathbf{a}^{(k)}$
6:    **return** $d, \mathbf{a}$
7: **procedure** DYNAMICPROGRAMMING($\mathbf{z}^{*(k)}, \hat{\mathbf{z}}^{(k)}$)
8:    $\hat{I} \leftarrow |\hat{\mathbf{z}}^{(k)}|; I^* \leftarrow |\mathbf{z}^{*(k)}|$            $\triangleright \hat{\mathbf{z}}^{(k)} = \hat{t}_1, \ldots, \hat{t}_{\hat{I}}$ and $\mathbf{z}^{*(k)} = t_1^*, \ldots, t_{I^*}^*$
9:    $\mathbf{D} \leftarrow$ zero matrix with $(M+1)$ rows and $(N+1)$ columns
10:   $\mathbf{P} \leftarrow$ empty matrix with $M$ rows and $N$ columns        $\triangleright$ *back pointers*
11:   **for** $\hat{i} \leftarrow 1$ **to** $\hat{I}$ :         $\triangleright$ *transport reference of length 0 to proposal of length $\hat{i}$*
12:     $\mathbf{D}_{\hat{i},0} \leftarrow \mathbf{D}_{\hat{i}-1,0} + C_{\text{insert}}$      $\triangleright$ *insert $t_i^* = \hat{t}_{\hat{i}}$ to reference (and their prefixes are matched)*
13:   **for** $i^* \leftarrow 1$ **to** $I^*$ :        $\triangleright$ *transport preference of length $i^*$ to proposal of length 0*
14:     $\mathbf{D}_{0,j^*} \leftarrow \mathbf{D}_{0,j^*} + C_{\text{delete}}$        $\triangleright$ *delete $t_{i^*}^*$ (and prefixes are matched)*
15:   **for** $\hat{i} \leftarrow 1$ **to** $\hat{I}$ :            $\triangleright$ *proposal prefix of length $\hat{i}$*
16:     **for** $i^* \leftarrow 1$ **to** $I^*$ :       $\triangleright$ *to match reference of length $i^*$*
17:       $D_{\text{insert}} \leftarrow \mathbf{D}_{\hat{i}-1,i^*} + C_{\text{insert}}$    $\triangleright$ *if an event token at $\hat{t}_{\hat{i}}$ is inserted to $\mathbf{z}^{*(k)}$*
18:       $D_{\text{delete}} \leftarrow \mathbf{D}_{\hat{i},i^*-1} + C_{\text{delete}}$    $\triangleright$ *if the event token at $t_{i^*}^*$ is deleted from $\mathbf{z}^{*(k)}$*
19:       $D_{\text{move}} \leftarrow \mathbf{D}_{\hat{i}-1,i^*-1} + |\hat{t}_{\hat{i}} - t_{i^*}^*|$   $\triangleright$ *if the event at $t_{i^*}^*$ of $\mathbf{z}^{*(k)}$ is aligned to event at $\hat{t}_{\hat{i}}$ of $\hat{\mathbf{z}}^{(k)}$*
20:       $\mathbf{D}_{\hat{i},i^*} \leftarrow \min\{D_{\text{insert}}, D_{\text{delete}}, D_{\text{move}}\}$    $\triangleright$ *choose the edit that yields the shortest distance*
21:       $\mathbf{P}_{\hat{i},i^*} \leftarrow \arg\min_{e \in \{\text{insert,delete,move}\}} D_e$    $\triangleright$ *e represents a kind of edition*
22:   $\hat{i} \leftarrow \hat{I}; i^* \leftarrow I^*; \mathbf{a} \leftarrow$ empty collection$\{\}$
23:   **while** $\hat{i} > 0$ **and** $i^* > 0$ :                 $\triangleright$ *back trace*
24:     **if** $\mathbf{P}_{\hat{i},i^*} =$ insert :             $\triangleright$ *token $t_{i^*}^*$ is deleted.*
25:       $\hat{i} \leftarrow \hat{i} - 1$
26:     **if** $\mathbf{P}_{\hat{i},i^*} =$ delete :            $\triangleright$ *a token at $\hat{t}_{\hat{i}}$ is inserted*
27:       $i^* \leftarrow i^* - 1$
28:     **if** $\mathbf{P}_{\hat{i},i^*} =$ move :             $\triangleright$ *token $t_{i^*}^*$ is aligned to $\hat{t}_{\hat{i}}$*
29:       $\hat{i} \leftarrow \hat{i} - 1; i^* \leftarrow i^* - 1$
30:       $\mathbf{a} \leftarrow \mathbf{a} \cup \{(\hat{t}_{\hat{i}}, t_{i^*}^*)\}$
31:   **return** $\mathbf{D}_{\hat{I},I^*}, \mathbf{a}$

---

they are already matched without any operations, $\mathbf{z}_1$ and $\mathbf{z}_2$ must be identical, thus the left direction of equation (22) holds.

OTD is symmetric, that is, $D(\mathbf{z}_1, \mathbf{z}_2) = D(\mathbf{z}_2, \mathbf{z}_1)$, if we set $C_{\text{insert}} = C_{\text{delete}}$. Suppose that $\mathbf{a}$ is an alignment between $\mathbf{z}_1$ and $\mathbf{z}_2$. It's easy to see that the only difference between $D(\mathbf{z}_1, \mathbf{z}_2, \mathbf{a})$ and $D(\mathbf{z}_2, \mathbf{z}_1, \mathbf{a})$ [8] is that the insertion and deletion operations are exchanged. For example, if we delete a token $t_i \in \mathbf{z}_1$ when calculating $D(\mathbf{z}_1, \mathbf{z}_2, \mathbf{a})$, we should insert a token at $t_i$ to $\mathbf{z}_2$ when calculating $D(\mathbf{z}_2, \mathbf{z}_1, \mathbf{a})$. If we set $C_{\text{insert}} = C_{\text{delete}}$, we have

$$D(\mathbf{z}_1, \mathbf{z}_2, \mathbf{a}) = D(\mathbf{z}_2, \mathbf{z}_1, \mathbf{a}), \quad \forall \mathbf{a} \in \mathcal{A}(\mathbf{z}_1, \mathbf{z}_2). \tag{23}$$

Therefore, we could obtain

$$D(\mathbf{z}_1, \mathbf{z}_2) = \min_{\mathbf{a}^* \in \mathcal{A}(\mathbf{z}_1, \mathbf{z}_2)} D(\mathbf{z}_1, \mathbf{z}_2, \mathbf{a}^*) = \min_{\mathbf{a}^* \in \mathcal{A}(\mathbf{z}_1, \mathbf{z}_2)} D(\mathbf{z}_2, \mathbf{z}_1, \mathbf{a}^*) = D(\mathbf{z}_2, \mathbf{z}_1). \tag{24}$$

Finally let's prove that OTD satisfies triangle inequality, that is:

$$D(\mathbf{z}_1, \mathbf{z}_2) + D(\mathbf{z}_2, \mathbf{z}_3) \geq D(\mathbf{z}_1, \mathbf{z}_3), \tag{25}$$

where $\mathbf{z}_1$, $\mathbf{z}_2$ and $\mathbf{z}_3$ are three sequences. This property could be proved intuitively. Suppose that the operations on $\mathbf{z}_1$ with minimal costs to make $\mathbf{z}_1$ matched to $\mathbf{z}_2$ are denoted by $o_1, o_2, \ldots, o_{n_1}$, and

---

[8] We abuse the notation $\mathbf{a}$, which we think could represent both the movement from $\mathbf{z}_1$ to $\mathbf{z}_2$ and from $\mathbf{z}_2$ to $\mathbf{z}_1$.

those on $\mathbf{z}_2$ to make $\mathbf{z}_2$ matched to $\mathbf{z}_3$ are denoted by $o'_1, o'_2, \ldots, o'_{n_2}$. $o_i$ could be a deletion, insertion or movement on a token. To make $\mathbf{z}_1$ matched to $\mathbf{z}_3$, one possible way, which is not necessarily the optimal, is to do $o_1, o_2, \ldots, o_{n_1}, o'_1, o'_2, \ldots, o'_{n_2}$ on $\mathbf{z}_1$. Since the total cost is the accumulation of the cost of each operation, and the operations on $\mathbf{z}_1$ above to make $\mathbf{z}_1$ matched to $\mathbf{z}_3$ might not be optimal, the triangle inequality equation (25) holds.

## D    APPROXIMATE MBR DETAILS

In this section, we first prove the Theorem 1 in section 4.2, and then show the detailed algorithm to find the decode.

### D.1    THEOREM PROOF AND RELATED

We have a claim in section 4.2 that:

**Theorem 1.** *Given* $\{\mathbf{z}_m\}_{m=1}^M$, *if we define* $\mathbf{z}_\sqcup = \bigsqcup_{m=1}^M \mathbf{z}_m$, *then:*

$$\exists \hat{\mathbf{z}} \in \mathcal{P}(\mathbf{z}_\sqcup) \text{ such that } \sum_{m=1}^M w_m D(\mathbf{z}_m, \hat{\mathbf{z}}) = \min_{\mathbf{z} \in \mathcal{Z}} \sum_{m=1}^M w_m D(\mathbf{z}_m, \mathbf{z}) \tag{26}$$

*where* $\mathcal{P}(\mathbf{z})$ *is the power set of any given* $\mathbf{z}$—*the set of subsequences of* $\mathbf{z}$ *(including empty sequence and* $\mathbf{z}$ *itself). That is to say, there exists one subsequence of* $\mathbf{z}_\sqcup$ *that achieves the minimum Bayes risk (or the minimum weighted optimal transport distance).*

and we prove it in this section:

*Proof.* Here we assume that there is only one type of event. Since the distances of different types of events are calculated separately, our conclusion is easy to be extended to the general case.

Suppose $\mathbf{z}^*$ is an optimal decode, that is,

$$\sum_{m=1}^M w_m D(\mathbf{z}_m, \mathbf{z}^*) = \min_{\mathbf{z} \in \mathcal{Z}} \sum_{m=1}^M w_m D(\mathbf{z}_m, \mathbf{z}).$$

If $\mathbf{z}^* \in \mathcal{P}(\mathbf{z}_\sqcup)$, the proof is done. If not, suppose there exists a token at $t_i \notin \mathbf{z}_\sqcup$. We use $t_l \in \mathbf{z}_\sqcup$ ($t_r \in \mathbf{z}_\sqcup$) to denote the token in $\mathbf{z}_\sqcup$ that is left (right) and nearest to $t_i$.[9] We will show that if we move $t_i$ around, as long as $t_i \in [t_l, t_r]$, the weighted optimal transport distance, i.e. $\sum_{m=1}^M w_m D(\mathbf{z}_m, \mathbf{z}^*)$, will neither increase nor decrease.

Suppose $\mathbf{a}_m^* = \arg\min_{\mathbf{a}_m \in \mathcal{A}(\mathbf{z}_m, \mathbf{z}^*)} \sum_{m=1}^M w_m D(\mathbf{z}_m, \mathbf{z}^*, \mathbf{a}_m)$. Let's use $r(t)$ to indicate the weighted transport distance of $\mathbf{z}^*$ with fixed alignment if we move $t_i$ to $t$, that is,

$$r(t) \stackrel{\text{def}}{=} \sum_{m=1}^M w_m D(\mathbf{z}_m, \mathbf{z}^*(t), \mathbf{a}_m^*),$$

where $\mathbf{z}^*(t)$ is the sequence $\mathbf{z}^*$ with $t_i$ moved to $t$. Because $\mathbf{z}^*(t_i)$ is an optimal decode, and $\mathbf{a}_m^*$ is the optimal alignment for $\mathbf{z}^*(t_i)$, we should have

$$r(t_i) = \min_t r(t).$$

Note that the transport distance is comprised of three parts: deletion, insertion and alignment costs. Since every $\mathbf{a}_m^*$ is fixed, if we change $t$, only the alignment cost that related to token $t$ will affect $r(t)$. This part of $r(t)$ is linear to $t$, since we have a constraint $t \in [t_l, t_r]$, which guarantees that it will not cross any other tokens in $\mathbf{z}_\sqcup$.

Since $r(t)$ is linear to $t \in [r_l, t_r]$ and $r(t)$ gets minimized at $t_i \in (t_l, t_r)$, we could conclude that

$$r(t) = r(t_i) = \text{Const}, \forall t \in [t_l, t_r].$$

---

[9]We assume that $t_i$ is in between two tokens in $\mathbf{z}_\sqcup$, and our proof could be easily extended to the case for which this condition is not satisfied.

| DATASET | $K$ | # OF EVENT TOKENS | | | SEQUENCE LENGTH | | |
|---------|-----|-------|-----|------|-----|------|-----|
| | | TRAIN | DEV | TEST | MIN | MEAN | MAX |
| SYNTHETIC | 4 | $\approx 74967$ | $\approx 7513$ | $\approx 7507$ | 10 | $\approx 15$ | 20 |
| NYCTAXI | 10 | 157916 | 15826 | 15808 | 22 | 32 | 38 |
| ELEVATOR | 10 | 313043 | 31304 | 31206 | 235 | 313 | 370 |

Table 1: Statistics of each dataset. We write "$\approx N$" to indicate that $N$ is the average value over multiple datasets of one kind (synthetic); the variance is small in each such case.

Since $r(t)$ is the upper bound of the weighted optimal transport distance $\sum_{m=1}^{M} w_m D(\mathbf{z}_m, \mathbf{z}^*(t))$, which also gets the same minimal value at $t_i \in (t_l, t_r)$ as $r(t)$, we could conclude that

$$\sum_{m=1}^{M} w_m D(\mathbf{z}_m, \mathbf{z}^*(t)) = \sum_{m=1}^{M} w_m D(\mathbf{z}_m, \mathbf{z}^*(t_i)) = \text{Const}, \forall t \in [t_l, t_r].$$

Therefore we could move token $t_i$ to either $t_l$ or $t_r$ without increasing the Bayes risk. We could do this movement for each $t_i \notin \mathbf{z}_\sqcup$ to get a new decode $\hat{\mathbf{z}} \in \mathcal{P}(\mathbf{z}_\sqcup)$, which is also an optimal decode.

$\square$

## D.2 ALGORITHM DETAILS

The detailed algorithm is presented in Algorithm 3.

## E EXPERIMENTAL DETAILS

In this section, we elaborate on the details of data generation, processing, and experimental results.

In all of our experiments, the distribution $p$ is trained on the complete (uncensored) version of the training data. The system is then asked to complete the incomplete (censored) version of the test (or dev) data. For particle smoothing, the proposal distribution is trained using both the complete and incomplete versions of the training data, as explained at the end of section 3.2.1.

In each experiment, the missingness mechanism is defined to be as

$$p_{\text{miss}}(Miss = \mathbf{z} \mid Comp = \mathbf{x} \sqcup \mathbf{z}) = \prod_{k_i @ t_i \in \mathbf{z}} \rho_{k_i} \prod_{k_i @ t_i \in \mathbf{x}} (1 - \rho_{k_i}), \tag{27}$$

meaning that each event in the complete stream $\mathbf{x} \sqcup \mathbf{z}$ is independently censored with probability $\rho_k$ that only depends on its event type $k$. Our experiments on imputing missing data assume that both the missingness mechanism equation (27) and its parameter vector $\boldsymbol{\rho}$ are known, although Appendix F discusses how $\boldsymbol{\rho}$ could be imputed when complete and incomplete data are both available..

For our main experiments in section 5.2, $\rho_k$ is either 0 or 1, which means some event types are always observed, while others are always missing. Though this missingness mechanism is deterministic, it can still be regarded as MNAR, because the factor $p_{\text{miss}}(Miss = \mathbf{z} \mid Comp = \mathbf{x} \sqcup \mathbf{z})$ in equation (3) is not constant. The factor is 1 if $\mathbf{z}$ consists of precisely the events in $\mathbf{x} \sqcup \mathbf{z}$ that ought to go missing, and 0 otherwise. In other words, $p_{\text{miss}}$ here ensures only that $\mathbf{z}$ should consist only of events with $\rho_k > 0$. Recall from Appendix A that $p_{\text{miss}}$ is also considered by the proposal distribution.

We also experiment with a more typical MNAR setting in Appendix G below, in which events are stochastically missing rather than deterministically missing.

## E.1 DATASET STATISTICS

Table 1 shows statistics about each dataset that we use in this paper.

## E.2 TRAINING DETAILS

We used single-layer LSTMs (Hochreiter & Schmidhuber, 1997), selected the number $D$ of hidden nodes of the left-to-right LSTM, and then $D'$ of the right-to-left one from a small set

$\{16, 32, 64, 128, 256, 512, 1024\}$ based on the performance on the dev set of each dataset. The best-performing $(D, D')$ pairs are $(256, 128)$ on Synthetic, $(256, 256)$ on Elevator $(256, 256)$ on NYC Taxi, but we empirically found that the model performance is robust to these hyperparameters. For the chosen $(D, D')$ pair on each dataset, we selected $\beta$ based on the performance on the dev set, and $\beta = 1.0$ yields the best performance across all the datasets we use. For learning, we used Adam with its default settings (Kingma & Ba, 2015).

### E.3 SYNTHETIC DATASETS DETAILS

Each of the ten neural Hawkes processes has its parameters sampled from $\mathrm{Unif}[-1.0, 1.0]$. Then a set of event sequences is drawn from each of them via the plain vanilla thinning algorithm (Mei & Eisner, 2017). For each of the ten synthetic datasets, we took $K = 4$ as the number of event types. To draw each event sequence, we first chose the sequence length $I$ (number of event tokens) uniformly from $\{11, 12, \ldots, 20\}$ and then used the thinning algorithm to sample the first $I$ events over the interval $[0, \infty)$. For subsequent training or testing, we treated this sequence (appropriately) as the complete set of events observed on the interval $[0, T]$ where $T = t_I$, the time of the last generated event.

We generate 5000, 500 and 500 sequences for each training, dev, and test set respectively. For the missingness mechanism, we censor all events of type 3 and 4. In other words, we set $\rho_k = 0$ for $k = 1, 2$ and $\rho_k = 1$ for $k = 3, 4$.

### E.4 ELEVATOR SYSTEM DATASET DETAILS

We examined our method in a simulated 5-floor building with 2 elevator cars. During a typical afternoon down-peak rush hour (when passengers go from floor-2,3,4,5 down to the lobby), elevator cars travel to each floor and pick up passengers that have (stochastically) arrived there according to a traffic profile (Bao et al., 1994). Each car will also avoid floors that already are or will soon be taken care of by the other. Having observed when and where car-1 has stopped (to pick up or drop off passengers) over this hour, we are interested in when and where car-2 has stopped during the same time period. In this dataset, each event type is a tuple of (car number, floor number) so there are $K = 10$ in total in this simulated 5-floor building with 2 elevator cars.

Passenger arrivals at each floor are assumed to follow a inhomogeneous Poisson process, with arrival rates that vary during the course of the day. The simulations we use follows a human-recorded traffic profile (Bao et al., 1994) which dictates arrival rates for every 5-minute interval during a typical afternoon down-peak rush hour. Table 2 shows the mean number of passengers (who are going to the lobby) arriving at floor-2,3,4,5 during each 5-minute interval.

We simulated the elevator behavior following a naive baseline strategy documented in Crites & Barto (1996).[10] In details, each car has a small set of primitive actions. If it is stopped at a floor, it must either "move up" or "move down". If it is in motion between floors, it must either "stop at the next floor" or "continue past the next floor". Due to passenger expectations, there are two constraints on these actions: a car cannot pass a floor if a passenger wants to get off there and cannot turn until it has serviced all the car buttons in its current direction. Three additional action constraints were made in an attempt to build in some primitive prior knowledge: 1) a car cannot stop at a floor unless someone wants to get on or off there; 2) it cannot stop to pick up passengers at a floor if another car is already stopped there; 3) given a choice between moving up and down, it should prefer moving up (since the down-peak traffic tends to push the cars toward the bottom of the building). Because of this last constraint, the only real choices left to each car are the stop and continue actions, and the baseline strategy always chooses to continue. The actions of the elevator cars are executed asynchronously since they may take different amounts of time to complete.

We repeated the (one-hour) simulation 700 times to collect the event sequences, each of which has around 300 time-stamped records of which car stops at which floor. We randomly sampled disjoint train, dev and test sets with 500, 100 and 100 sequences respectively.

We set $\rho_k = 0$ for $k = 1, \ldots, 5$ and $\rho_k = 1$ for $k = 6, \ldots, 10$, meaning that the events (of arriving at floor 1, 2, $\ldots$, 5) of car 1 are all observed, but those of car 2 are not.

---

[10]We rebuilt the system in Python following the original Fortran code of Crites & Barto (1996).

| Start Time (min) | 00 | 05 | 10 | 15 | 20 | 25 | 30 | 35 | 40 | 45 | 50 | 55 |
|---|---|---|---|---|---|---|---|---|---|---|---|---|
| Mean # Passenger | 1 | 2 | 4 | 4 | 18 | 12 | 8 | 7 | 18 | 5 | 3 | 2 |

Table 2: The Down-Peak Traffic Profile

### E.5 NEW YORK CITY TAXI DATASET DETAILS

The New York City Taxi dataset (section 5.1) includes 189,550 taxi pick-up and drop-off records in the city of New York in 2013. Each record has its medallion ID, driver license and time stamp. Each combination of medallion ID and driver license naturally forms a sequence of time-stamped pick-up and drop-off events. Following the processing recipe of previous work (Du et al., 2016), we construct shorter sequences by breaking each long sequence wherever the temporal gap between a drop-off event and its following pick-up event is larger than six hours. Since the schedule of different drivers might be very different, it's hard to set a natural $t_0$, that is, the time stamp of BOS.

We randomly sampled a month from 2013 and then randomly sampled disjoint train, dev and test sets with 5000, 500 and 500 sequences respectively from that month.

In this dataset, each event type is a tuple of (borough, action) where the action can be either pick-up or drop-off, so there are $K = 5 \times 2 = 10$ event types in total.

We set $\rho_1 = 0$ and $\rho_2 = 1$, which means that all drop-off events but no pick-up events are observed.

## F MONTE CARLO EM

We normally assume (section 3.2.1) that some complete sequences are available for training the neural Hawkes process models. If incomplete sequences are also available, our particle smoothing method can be used to (approximately) impute the missing events, which yields additional complete sequences for training. Indeed, if we are willing to make an MAR assumption (Little & Rubin, 1987),[11] then we can do imputation without modeling the missingness mechanism. Training on such imputed sequences is an instance of **Monte Carlo expectation-maximization (MCEM)** (Dempster et al., 1977; Wei & Tanner, 1990; McLachlan & Krishnan, 2007), with particle smoothing as the Monte Carlo E-step, and makes it possible to train with incomplete data only. In the more general MNAR scenario, we can extend the E-step to consider the not-at-random missingness mechanism (see equation (7b) below), but then we need both complete and incomplete sequences at training time in order to fit the parameters of the missingness mechanism (unless these parameters are already known) jointly with those of the neural Hawkes process. Although training with incomplete data is out of the scope of our experiments, we describe the methods and provide MCEM pseudocode in Appendix F.

In this case, we would like to know the probability of the observed data under the target distribution:

$$p(Obs = \mathbf{x}) = \sum_{\mathbf{z}} p(Miss = \mathbf{z}, Obs = \mathbf{x}) = \sum_{\mathbf{z}} p(\mathbf{x} \sqcup \mathbf{z}) p_{\text{miss}}(\mathbf{z} \mid \mathbf{x}) \tag{28}$$

where the marginal target distribution $p(Obs = \mathbf{x})$ is abbreviated as $p(\mathbf{x})$. If we propose $\mathbf{z}$ from $q(\mathbf{z} \mid \mathbf{x})$, then it can be rewritten as:

$$p(\mathbf{x}) = \sum_{\mathbf{z}} p(\mathbf{x} \sqcup \mathbf{z}) p_{\text{miss}}(\mathbf{z} \mid \mathbf{x}) \frac{q(\mathbf{z} \mid \mathbf{x})}{q(\mathbf{z} \mid \mathbf{x})} d\mathbf{z} = \mathbb{E}_{\mathbf{z} \sim q(\mathbf{z}|\mathbf{x})} \left[ \frac{p(\mathbf{x} \sqcup \mathbf{z}) p_{\text{miss}}(\mathbf{z} \mid \mathbf{x})}{q(\mathbf{z} \mid \mathbf{x})} \right] \tag{29}$$

Given a finite number $M$ of proposed particles $\{\mathbf{z}_m\}_{m=1}^M$, this expectation can be estimated with empirical average:

$$p(\mathbf{x}) = \frac{1}{M} \sum_{m=1}^M \frac{p(\mathbf{x} \sqcup \mathbf{z}_m) p_{\text{miss}}(\mathbf{z}_m \mid \mathbf{x})}{q(\mathbf{z}_m \mid \mathbf{x})} \tag{30}$$

and it is obvious that

$$\log p(\mathbf{x}) \geq \frac{1}{M} \sum_{m=1}^M (\log p(\mathbf{x} \sqcup \mathbf{z}_m) + \log p_{\text{miss}}(\mathbf{z}_m \mid \mathbf{x}) - \log q(\mathbf{z}_m \mid \mathbf{x})) \tag{31}$$

---

[11]It is "almost impossible" to determine from the data whether the MAR assumption holds (Mohan & Pearl, 2018).

where the right-hand-side (RHS) term is the **Evidence Lower Bound (ELBO)** that we would maximize in order to maximize the log-likelihood.

The MCEM algorithm is composed of two steps:

**E(xpectation)-step** We train the proposal distribution $q(\mathbf{z} \mid \mathbf{x})$ using the method in section 3.2.1 and then sample $M$ weighted particles from $q(\mathbf{z} \mid \mathbf{x})$ by calling Algorithm 1.

**M(aximization)-step** We train the neural Hawkes process $p(\mathbf{x} \sqcup \mathbf{z})$ by maximizing the ELBO (equation (31)).

Note that in the MAR case, $p_{\mathrm{miss}}(\mathbf{z} \mid \mathbf{x})$ is constant of $\mathbf{z}$ so the it can be omitted from the formulation (and thus the algorithms). Also note that, for particle filtering, the proposal distribution $q(\mathbf{z} \mid \mathbf{x})$ is only part of $p(\mathbf{x} \sqcup \mathbf{z})$ so we ought only to propose particles in the E-step.

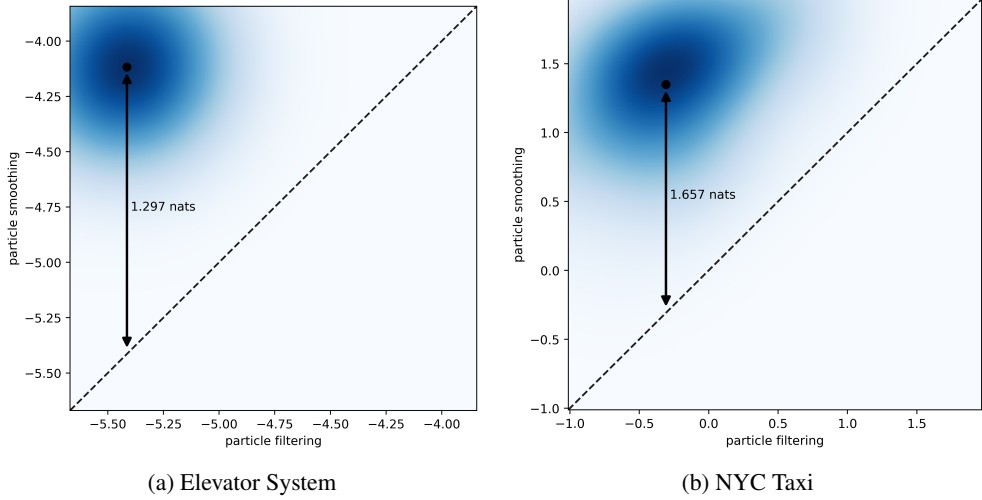

(a) Elevator System

(b) NYC Taxi

Figure 4: Scatterplots with $\rho = 0.5$. Same comparison as Figure 2.

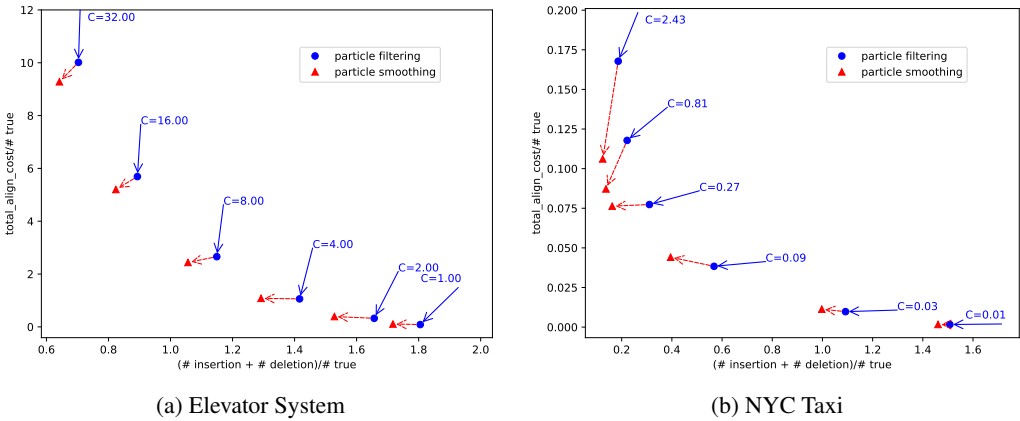

(a) Elevator System

(b) NYC Taxi

Figure 5: Optimal transport distance results with $\rho = 0.5$. Same comparison as Figure 3.

## G  MORE EXPERIMENTS ON MNAR DATA

Our experimental setting in section 5.1 and Appendix E is a special case of MNAR where some event types are deterministically censored. In this section, we consider a more typical MNAR setting, where (in the notation of Appendix E) we set $\rho_k = \rho = 0.5$ regardless of the event type $k$. Then equation (27) can be written as $p_{\mathrm{miss}}(Miss = \mathbf{z} \mid Comp = \mathbf{x} \sqcup \mathbf{z}) = (0.5)^{|\mathbf{x}|+|\mathbf{z}|}$, whose value depends on the length of $|\mathbf{z}|$. This is obviously non-constant over $\mathbf{z}$.

The proposal distribution in this case will be more conservative about proposing missing events, because having a lot of missing events is *a posteriori* improbable. In other words, $p_{\mathrm{miss}}$ as given above falls off with the number of missing events $|\mathbf{z}|$.

We conducted experiments on the real-world datasets in this setting. Again, the method works, with very similar qualitative behavior to before. As shown in Figures 4 and 5, the particle smoothing method outperforms the filtering baseline (i.e. the neural version of Linderman et al. (2017)). Note that these figures look very similar to Figures 2 and 3. This is not too surprising because these two settings both have about half their events censored—thus being about equally difficult to reconstruct.

---

**Algorithm 3** Find an approximately minimum-Bayes-risk sequence of events

---

**Input:** collection of weighted particles $\mathcal{Z}_M = \{(\mathbf{z}_m, w_m)\}_{m=1}^M$
**Output:** decode $\hat{\mathbf{z}}$

1: **procedure** APPROXMBR($\mathcal{Z}_M$)
2:   $\hat{\mathbf{z}} \leftarrow$ empty sequence
3:   **for** $k = 1$ **to** $K$ :
4:     $\hat{\mathbf{z}}^{(k)} \leftarrow$ DECODEK($\{(\mathbf{z}_m^{(k)}, w_m)\}_{m=1}^M$)              ▷ *decode for type-k by calling* DECODEK
5:     $\hat{\mathbf{z}} \leftarrow \hat{\mathbf{z}} \sqcup \hat{\mathbf{z}}^{(k)}$
6:   **return** $\hat{\mathbf{z}}$
7: **procedure** DECODEK($\mathcal{Z}_M$)
8:   ▷ $\mathcal{Z}_M$ *actually means* $\mathcal{Z}_M^{(k)} = \{(\mathbf{z}_m^{(k)}, w_m)\}_{m=1}^M$ *throughout the procedure;* $\mathbf{z}_m$ *is constant*
9:   $\mathbf{z} \leftarrow \arg\max_{\mathbf{z} \in \{\mathbf{z}_m\}_{m=1}^M} w_m$              ▷ *init decode as highest weighted particle and it is global*
10:   **repeat**
11:     **for** $m = 1$ **to** $M$ :                                   ▷ **Align Phase**
12:       $d_m, \mathbf{a}_m \leftarrow$ DYNAMICPROGRAM($\mathbf{z}_m, \mathbf{z}$)     ▷ *call method in Algorithm 2;* $d_m, \mathbf{a}_m$ *are global*
13:     $r_{\min} \leftarrow \sum_m w_m d_m$                               ▷ *track the risk of current* $\mathbf{z}$
14:     $\mathbf{z}, \{d_m, \mathbf{a}_m\}_{m=1}^M \leftarrow$ MOVE($\mathbf{z}, \{\mathbf{z}_m, d_m, \mathbf{a}_m\}_{m=1}^M$)
15:     $\mathbf{z}, \{d_m, \mathbf{a}_m\}_{m=1}^M \leftarrow$ DELETE($\mathbf{z}, \{\mathbf{z}_m, d_m, \mathbf{a}_m\}_{m=1}^M$)
16:     $\mathbf{z}, \{d_m, \mathbf{a}_m\}_{m=1}^M \leftarrow$ INSERT($\mathbf{z}, \{\mathbf{z}_m, d_m, \mathbf{a}_m\}_{m=1}^M$)
17:   **until** $\sum_{m=1}^M w_m d_m = r_{\min}$                          ▷ *risk stops decreasing*
18:   **return** $\mathbf{z}$
19: **procedure** MOVE($\mathbf{z}, \{\mathbf{z}_m, d_m, \mathbf{a}_m\}_{m=1}^M$)                  ▷ **Move Phase**
20:   **for** $t$ **in** $\mathbf{z}$ :
21:     **for** $t' \in \{t' : (t', t) \in \bigcup_{m=1}^M \mathbf{a}_m\}$ :       ▷ *may replace t with t' which is aligned to t*
22:       $(\forall m) d_m' \leftarrow d_m$
23:       **for** $(t'', m) \in \{(t'', m) : (t'', t) \in \mathbf{a}_m, m \in \{1, \ldots, M\}\}$ :
24:         $d_m' \leftarrow d_m' - |t'' - t| + |t'' - t'|$
25:       **if** $\sum_m w_m d_m' < \sum_m w_m d_m$ :
26:         $(\forall m) d_m \leftarrow d_m'; t \leftarrow t'$                       ▷ *t move to t' for lower risk*
27:   **return** $\mathbf{z}, \{d_m, \mathbf{a}_m\}_{m=1}^M$
28: **procedure** DELETE($\mathbf{z}, \{\mathbf{z}_m, d_m, \mathbf{a}_m\}_{m=1}^M$)                  ▷ **Delete Phase**
29:   **for** $t$ **in** $\mathbf{z}$ :                                        ▷ *may delete this event*
30:     **for** $m = 1$ **to** $M$ :                                      ▷ *update each* $d_m$
31:       **if** $\exists t' \in \mathbf{z}_m$ **and** $(t', t) \in \mathbf{a}_m$ :       ▷ *find the only, if any,* $t' \in \mathbf{z}_m$ *that is aligned to t*
32:         $d_m' \leftarrow d_m + C_{\text{delete}} - |t' - t|$              ▷ $d_m$ *changes if we delete t and alignment* $(t', t)$
33:       **else**                                            ▷ *otherwise, one insertion must have been made to match t*
34:         $d_m' \leftarrow d_m - C_{\text{insert}}$                     ▷ *so the insertion cost disappears if we delete t*
35:       **if** $\sum_m w_m d_m' < \sum_m w_m d_m$ :
36:         delete $t$ from $\mathbf{z}$; $(\forall m)$ delete $(t', t)$ from $\mathbf{a}_m$; $d_m \leftarrow d_m'$
37:   **return** $\mathbf{z}, \{d_m, \mathbf{a}_m\}_{m=1}^M$
38: **procedure** INSERT($\mathbf{z}, \{\mathbf{z}_m, d_m, \mathbf{a}_m\}_{m=1}^M$)                  ▷ **Insert Phase**
39:   **for** $t \in \{t : t \in \bigcup_{m=1}^M \mathbf{z}_m, t \notin \mathbf{z}\}$ :               ▷ *may insert t' if it is not in* $\mathbf{z}$ *yet*
40:     $(\forall m) \mathbf{a}_m' \leftarrow \mathbf{a}_m$
41:     **for** $m = 1$ **to** $M$ :          ▷ *find t' in* $\mathbf{z}_m$ *that is not aligned (thus in* $\mathbf{z}'$*) and closest to t* (arg min)
42:       $\mathbf{z}' \leftarrow \{t' : \forall t, (t', t) \notin \mathbf{a}_m\}$                    ▷ $\mathbf{z}'$ *may be empty, i.e. all in* $\mathbf{z}_m$ *are aligned*
43:       **if** $\mathbf{z}'$ is not empty **and** $|(t' \leftarrow \arg\min_{t' \in \mathbf{z}_m \cap \mathbf{z}'}\{|t' - t|\}) - t| < C_{\text{insert}} + C_{\text{delete}}$ :
44:         $d_m' \leftarrow d_m - C_{\text{delete}} + |t' - t|$; add $(t', t)$ to $\mathbf{a}_m'$
45:       **else**
46:         $d_m' \leftarrow d_m + C_{\text{insert}}$
47:     **if** $\sum_m w_m d_m' < \sum_m w_m d_m$ :
48:       insert $t$ to $\mathbf{z}$; $(\forall m) d_m \leftarrow d_m'; \mathbf{a}_m \leftarrow \mathbf{a}_m'$
49:   **return** $\mathbf{z}, \{d_m, \mathbf{a}_m\}_{m=1}^M$

---

