# OpenReview forum: "Inference of unobserved event streams with neural Hawkes particle smoothing"
_ICLR.cc/2019/Conference_

### Official Review · AnonReviewer1 · 2018-11-01
**Re: particle smoothing for neural Hawkes Processes**

**Rating:** 5
**Confidence:** 3

**Review:**

The paper presents an inference method (implicit distribution particle smoothing) for neural Hawkes processes that accounts for latent sequences of events that influence the observed trajectories.

Quality
+ The paper combines ideas from multiple areas of machine learning to tackle a challenging task of inference in multivariate continuous-time settings.
- The figures reported from the paper are comparative graphs with respect to particle filtering, and so the absolute level of performance of the methods is not characterized.  Reporting of distribution of sample weights and or run-times/complexity would strengthen the paper.

Clarity
- notation is complex replete with symbols "@" and text in math formulas
- It's not clear what p ("the data model") and p_miss ("the missingness mechanism") represent, and therefore why in equation 1: p(x,z) = p(xvz)p_miss(z| xvz) where v is the union symbol.  In addition, how it's related to MAR and MNAR is unclear. If e.g. following Murphy, one writes MAR as: p(r|x_u, x_o) = p(r|x_o), r is a missingness vector, x_u is x unobserved, and x_o is x observed, then r corresponds to observation or not, whereas in the manuscript p_miss is on the values themselves, i.e. on the space where z={k_{i,j}@t_{i,j}} resides.  We know, from the definition of MNAR that we can't use only the observed data to correctly infer the distributions of the missing values, and so while one can probabilistically predict in MNAR setting, their quality remains unknown.  If none of the experiments touch upon MNAR data, perhaps it is possible to omit this part.

Originality
+ the work is rich, complex, original, and uses leading methods from multiple areas of ML.

Significance
+ the significance of this work could be high, as it may provide a way to conduct difficult inference in an effective way to produce increasingly flexible modeling of trajectories amidst partial observation.
- however the exposition (particularly the experiments) does not fully demonstrate this.

---

> ### Author Response · Authors · 2018-11-19
> **Subject: MNAR data experiments (4/4)**
>
> > We know, from the definition of MNAR that we can't use only the observed data to correctly infer the distributions of the missing values, and so while one can probabilistically predict in MNAR setting, their quality remains unknown.
>
> Sure, working with MNAR data is impossible without additional knowledge. But in our setting, we have that additional knowledge.
>
> The problem with MNAR is that JOINTLY identifying p and p_miss is impossible. If you observe few 50-year-olds on your survey, you can't know (beyond your prior) whether that’s because there are few 50-year-olds, or because 50-year-olds are very likely to omit their age.
>
> But joint identification is unnecessary if either
> (1) one has separate knowledge of the missingness distribution p_miss
> (2) one has separate knowledge of the complete-data distribution p
>
> That is: If we know at least one of the distributions, then we can still infer the other. Actually, both (1) and (2) hold in our present experiments.
>
> The E step of EM uses the current guess of p and p_miss to infer the posterior distribution of the missing values. That posterior is uncontroversially defined by the simple Bayesian formula (3).
>
> (1) If p_miss is known and fixed, this gives a minor variant of ordinary EM. Ordinary EM makes the MAR assumption that the p_miss factor of (3) can be ignored. But we don't need to ignore p_miss if we actually know it! In our experiments, p_miss is MNAR but we do know it: we know that events of some types are always observed and events of other types are never observed. So, no problem!
>
> (2) Conversely, if p is known because we estimated it FROM SOME COMPLETE DATA, then we can use incomplete data to learn the MNAR missingness distribution p_miss. This setting even lets us learn a fancy missingness mechanism, e.g., some BiLSTM model that uses the context of an event to determine the probability of censoring it.
>
> We relegated this EM discussion to Appendix F since it is not used in our experiments. Appendix F says: “In the more general MNAR scenario, we can extend the E-step to consider the not-at-random missingness mechanism (see equation (7b) below), but then we need both complete and incomplete sequences at training time in order to fit the parameters of the missingness mechanism (unless these parameters are already known) jointly with those of the neural Hawkes process.  ... we describe the methods and provide MCEM pseudocode.”
>
> > If none of the experiments touch upon MNAR data, perhaps it is possible to omit this part.
>
> Alas (as we mentioned in the “presentation of MAR and MNAR” response), for missing data in event streams, nearly *every* setting is MNAR!  That is, the probability that z would be selected for censorship depends on the number and type of events in z. In particular, the second factor of (3) typically decreases exponentially in the number of missing events |z|, so it is not constant in z as required for MAR.
>
> In particular, our experimental setting is MNAR in a sense described at the end of this response. Because it happens to be a special case of MNAR, it would be possible through a notational trick to gloss over the MNAR issue and not call the reader’s attention to it. However, we thought this would be dangerous, so we would prefer to clarify this aspect of the exposition rather than deleting it. (We did relegate part of the discussion to an appendix.)
>
> Why dangerous?  We imagine that a reader might try to apply our method to a fairly simple situation where each event of type k has independent probability c_k of being censored. We fear that the reader might carelessly omit the p_miss factor if we don’t talk about it. However, that factor is necessary to avoid proposing too many missing events of those types that would NOT tend to be censored.
>
> E.g., proposing 100 missing events of type k means that p_miss includes a factor of c_k ^ 100. Thus, for c_k < 1 and especially for c_k << 1, the system should prefer -- other things equal -- to posit only 50 missing events. Intuitively, for 50 events to all have gone missing is not as improbable as for 100 events to have gone missing.
>
> It’s true that our reported experiments happen to have c_k = 1 (that is, events of type k are *deterministically* missing), so this exponential decay does not occur: c_k ^ 100 == c_k ^50. Nonetheless, to ensure that a future reader would handle the general case correctly, we prefer to give it some discussion. We can also add experiments with c_k = 0.5 to cement the expository point.
>
> Even our deterministic setting should still be regarded as MNAR, because c_k isn’t *always* 1 in our experiments. Rather, c_k = 1 or 0 depending on k. Thus, our p_miss factor can be either 1 or 0 (making us MNAR). More precisely, p_miss = 0 if z includes events of a type k that would never go missing. This is the technical reason that our code never proposes such events — as explained at the bottom of page 14.

---

> > ### Author Response · Authors · 2018-11-26
> > **added short appendix G with new MNAR experiments**
> >
> > We wrote:
> > > We can also add experiments with c_k = 0.5 to cement the expository point.
> >
> > We have now added these experiments, which constitute new Appendix G in the supplementary material.
> > In these experiments, events are missing stochastically rather than deterministically.
> > We find that the method still works and has the same qualitative behavior.
> >
> > (We have renamed the missingness probability c_k to \rho_k.  See the start of Appendix E for the notation.)
> >
> > This setting is definitely MNAR.  A naive reader might protest that it appears to be MCAR, because whether an event is missing always has probability 0.5, independent of the type of the event.  However, this is the subtle point at issue.  It is MNAR because as we noted in the previous comment, "the second factor of (3) ... decreases exponentially in the number of missing events |z|."  (If you're still not convinced, reread our “presentation of MAR and MNAR” response.  We will work this into the final version of the paper, of course.)

---

> ### Author Response · Authors · 2018-11-19
> **Subject: presentation of MAR and MNAR (3/4)**
>
>
> > In addition, how it's related to MAR and MNAR is unclear. If e.g. following Murphy, one writes MAR as …
>
> This is indeed a subtle point, one that we are proud to have handled correctly.  If you did not find our exposition clear, we will revise the camera-ready to lay out the issues more plainly.  Let’s start in this response.
>
> Little & Rubin’s MCAR/MAR/MNAR taxonomy was meant for graphical models.  (Murphy’s textbook just recapitulates this standard taxonomy.)  A graphical model has a fixed set of random variables, and the missingness mechanisms envisioned by Little & Rubin simply decide which of those variables to reveal.
>
> We could have chosen to formulate our model in these terms, by using uncountably many random variables K_t where t ranges over the set of times.  K_t = k if there is an event of type k at time t, and otherwise K_t = 0.  Then a missing event corresponds to an unobserved variable K_t with value > 0. Values of 0 are never observed because we are never told that an event did *not* happen at time t.   Some values > 0 are observed and some are not.  Since the missingness of K_t depends on whether K_t > 0, this setting is ordinarily MNAR.
>
> However, we prefer to formulate our model in terms of the finite sequences that are generated or read by our LSTMs.  This improves the notation later in the paper.
>
> From that point of view, unfortunately, the complete draws from p are not fixed-length vectors as in a graphical model: different draws from p can have different numbers of events.  This is why our notation does not use a simple “missingness vector” of fixed finite length as in the standard notation.  A missing event is not a case of a variable whose value is missing (e.g., an event of unknown type): we don’t even know whether the variable (event) exists in the first place!
>
> Yet our treatment of MAR is the correct generalization of Little & Rubin’s: namely, it’s the case in which the second factor of (3) can be ignored.  (The ability to ignore that factor is precisely why anyone cares about the MAR case!)  This is discussed around equation (3) and in Appendix F.

---

> ### Author Response · Authors · 2018-11-19
> **Subject: clarity of notation (2/4)**
>
>
> We worked really hard on the exposition, including months of tinkering with the writing and getting feedback from colleagues.  Perhaps the subject matter is difficult, but we really worked to make it as clear as we could, and we stand by our presentational choices.
>
> Your specific objections seem to be a matter of taste -- but please recognize that they were intended to *improve* clarity.  We’re happy to debate the best notation, but please don’t reject a technical paper on this basis?  It’s not as if our notation was careless or incomplete.  The use of “Comp”, “Obs”, and “Miss” as random variable names was supposed to be more mnemonic than C, O, and M.  The notation k@t denotes an event of type k at (“@”) time t.  This was supposed to be an improvement on Mei & Eisner’s <k,t> notation because it distinguishes this kind of ordered pair from other kinds of ordered pairs (similar to the use of sigils in programming languages); it was suggested by a colleague.
>
> > It's not clear what p ("the data model") and p_miss ("the missingness mechanism") represent, and therefore why in equation 1: p(x,z) = p(xvz)p_miss(z| xvz) where v is the union symbol.
>
> This is spelled out carefully at the start of section 2 and around equation (1).  The generative story has two steps.  First, the data model p generates a complete event sequence Comp.  Then p_miss decides which of these events get revealed to the user.
>
> Obs = x is the resulting subsequence of revealed (observed) events, and Miss = z is the subsequence of unrevealed (missing) events.  In other words, the missingness mechanism partitions Comp into Obs and Miss.  p(x,z) is the joint probability of getting a particular complete event sequence x v z  *and* partitioning it into x and z.  We particularly needed the notation x because x is the sequence that our particle smoother reads from right to left.

---

> ### Author Response · Authors · 2018-11-19
> **Subject: experimental evaluation (1/4)**
>
> Thanks for the suggestions:
>
> > The figures reported from the paper are comparative graphs with respect to particle filtering, and so the absolute level of performance of the methods is not characterized.
>
> Note that we do show absolute performance on our downstream task (namely, imputation of missing events), via the axis labels in Figure 3.  Figure 3 also shows the impact of particle smoothing on this downstream task.
>
> Our Figure 2 also shows “absolute performance” on the axes, measured as log q(z* | x) where q is the proposal distribution.  It’s true that these numbers are hard to interpret.  Ideally we would compare them to log p(z* | x), since that would be the ideal proposal distribution.  But unfortunately it is intractable to compute that conditional probability: even for synthetic data we are only able to compute the joint probability p(x, z*).  Do you have any ideas?
>
> > Reporting of distribution of sample weights and or run-times/complexity would strengthen the paper.
>
> Regarding sample weights, we can report the effective sample size in the final version.  Our effective sample size is excellent for the synthetic datasets, about 20-25 for M=50 particles. On real datasets, ESS increased roughly as sqrt(M) as we varied M from 20 to 2000 in pilot experiments.  Unfortunately it was very low for M=50, the value of M that we used in our final experiments (ESS of 1.5 to 2.2 on average), yet we still got good imputation results on our task.  We might be able to raise the ESS by combining our method with multinomial resampling or local search.
>
> Regarding theoretical and wall-clock runtime, please see our separate response “Subject: efficiency of Monte Carlo methods”.  The TL;DR is that the runtime complexity is O(MI) where M is the number of particles and I is the number of observed events.  In practice, we generate the particles in parallel, leading to acceptable speeds of 300-400ms per event for the final method.  We can add this information to the final version of the paper.

---

### Official Review · AnonReviewer3 · 2018-11-04
**Interesting problem with weak experimental evaluation**

**Rating:** 4
**Confidence:** 5

**Review:**

The authors propose a particle smoothing approach with an approximate minimum Bayes risk decoder to impute missing events in the Neural Hawkes Process (NHP). The main goal is to address the missing events problem in continuous-time event analysis, which is an important problem in practice. The core idea is within the framework of particle smoothing.

To formulate the posterior distribution of the missing event, the authors consider both the left-to-right past events and the right-to-left future events. The paper first applies the NHP to capture both the observed and inferred missing events to learn a representation of the past events, and then uses a similar NHP to learn the representation of the observed events from the future. Based on the two representations, it then formulates the intensity function of the missing events and uses the thinning algorithm to sample different particles. Based on the proposed distribution, the paper also considers to decode a single prediction achieving the Minimum Bayes Risk. Experiments on synthetic datasets with 10 different initializations and two real datasets show that the proposed smoothing approach is better than the filtering baseline.

In general, this paper considers an important problem which is under active research in literature recently. However, there are a few weaknesses of the paper that should be addressed.

1. The proposed technique is tightly connected to NHP, which could limit the applicability of the approach to other temporal point processes. The essential idea is similar to Bi-LSTM to learn the representation from both ends of a sequence of asynchronous temporal events. There are several different ways to represent the inter-event time to feed into the network other than NHP. Can the proposed method also be applied to other processes?

2. Within the particle filtering framework, each particle (hypothesis) is weighted by the likelihood of the sequence of observed events under that hypothesis. It turns out that the integral part of Equation 5 does not have an obvious analytical solution under NHP. Then, we first need a set of samples to approximate the likelihood evaluation. Later, we also need to sample particles. I am not quite convinced the computational efficiency of this approach in real applications of practice. Also, there is no analysis either empirically or analytically about the impact of the accumulative sampling errors on the inference performance. Furthermore, to learn the proposed distribution, the paper applies the REINFORCE algorithm under the proposed distribution q. But REINFORCE is known for large variance issue. Given that we already need lots of samples for the likelihood, it is unclear to me how stable the algorithm could be in practice.

3. The experimental evaluation is weak. For particle filtering and smoothing, it is known that the filtering techniques are candidates for solving the smoothing problem but perform poorly when T is large. That's why it is necessary to develop more sophisticated strategies for good smoothing
algorithms. As a result, it is unfair to only compare the smoothing approach with the filtering baseline.

Actually, what people really care about is how different techniques can behave in real data to impute realistic missing events. From this perspective, I suggest to use the QQ-plot to evaluate the goodness of fitting on the synthetic dataset. For example, given a sequence of events generated from an independent temporal point process, we can randomly delete events, and then apply different techniques, including Linderman et al. (2017), Shelton et al.(2018), to impute missing events. Finally, we can compare the imputed sequence of events with the groundtruth.

In addition, sequential monte carlo approach often suffers from skewed particle issue where one particle gradually dominates all the other particles with no diversity. It is unclear how the proposed approach is able to handle this.

One missing related paper is "Learning Hawkes Processes from Short Doubly-Censored Event Sequences"

Section 5.2 can be significantly strengthened if comparing with at least one of these approaches.

4. The paper is fairly written. I had some trouble reading back and forth for understanding Figure 1 since it has long caption that is not self-contained. The annotation of Section 2 is also too heavy to quickly skim through to memorize.

---

> ### Author Response · Authors · 2018-11-19
> **Subject: experimental evaluation (5/5)**
>
>
> >  it is unfair to only compare the smoothing approach with the filtering baseline.
>
> Well, what other baseline do you think we should compare with?  There is not a lot of previous work on this problem.
>
> We can see that Metropolis-Hastings would be a possible alternative, where the transition kernel proposes a single-event change (insert, delete, or move).  Unfortunately, this would be quite slow for a neural model like ours.  The reason is that a proposed change early in the sequence will affect the LSTM state and hence the probability of all subsequent events.  Thus, a single move takes O(length of proposed complete sequence) time to evaluate.  Furthermore the Markov chain may mix slowly because a move that changes only one event may often lead to an incoherent sequence that will be rejected.  The point of particle smoothing is essentially to avoid this kind of rejection by proposing a *coherent sequence of events* from an approximation q to the true posterior.  We can ensure that it is coherent because we build it up from left to right (taking the future into account).
>
> We’d be happy of course to propose Metropolis-Hastings as future work.  It could even build on our present work by using a variant of our current proposal distribution as the core of a Metropolis-Hastings kernel -- which would resample the latent events on a given *interval*.  However, we would be wary of developing this nontrivial extension within the current paper; it is not an established baseline and would take a few additional pages to develop.  The current submission already has too much material -- there are a lot of appendices, and the other reviewers seem to have found the submission to be overwhelming already.
>
> Another good piece of future work would be particle Gibbs or other particle MCMC algorithms, which would also build on our present work.
>
> > sequential monte carlo approach often suffers from skewed particle issue where one particle gradually dominates all the other particles with no diversity.
>
> This is indeed a danger in SMC approaches.  But surely you don’t think that all SMC papers should be rejected just because they use SMC?   There are several techniques in the SMC community for “rejuvenating” a skewed ensemble, such as multinomial resampling, other forms of resampling, and the “particle cascade.”  Any of these techniques could be combined with ours, and this is orthogonal to the technical contributions of our paper.
>
> > It is unclear how the proposed approach is able to handle this.
>
> In fact, our particle smoothing method is also intended to alleviate this issue.  As you know, if we could achieve a perfect proposal distribution q that was proportional to p, then the particle weight p/q would be constant across all particles, completely eliminating the skew issue.  So our paper shows how to improve the proposal distribution.
>
> Specifically, the reason that an SMC ensemble becomes skewed over time is that some of the proposed particles turn out to be less compatible with the future, and are reweighted to have a weight near 0.  Particle smoothing tries to incorporate the future into the proposal distribution so that this will not happen as badly.
>
> > what people really care about is how different techniques can behave in real data to impute realistic missing events.
>
> We certainly agree!  Which is why our section 4 (backed by appendices C-D, including Algorithm 2) gives a sophisticated method for doing exactly that.  Results from applying this method to impute missing events on real data are reported in section 5.2, including the carefully designed Figure 3.
>
> Could you please reread that material, and raise your score as appropriate to recognize the work that we did there?
>
> You suggest Linderman et al. (2017) and Shelton et al. (2018) as if they would be appropriate baselines for this imputation task.  However, those papers only apply to Hawkes processes.  Please note that we did discuss them carefully in section 6.
>
> (Specifically: Our particle filtering baseline is already the SAME as Linderman et al. (2017), just extended from the Hawkes process to the *neural* Hawkes process.  Shelton et al. (2018) use MCMC, but their MCMC algorithm takes advantage of special properties of the Hawkes process.  Unfortunately, those special properties no longer hold for the *neural* Hawkes process, which would therefore require a much slower MCMC algorithm, as noted above; we haven’t tried that.)
>
> (You also suggest that Xu et al. (2017) is relevant.  We are happy to cite it in the final version, but note that that paper focuses on quite a different kind of missing data -- “short reads” where a long sequence has been broken up and it is not known which pieces go together.  The first author of that paper agreed that his paper isn’t directly comparable to our setting, when we corresponded with him before submission.)

---

> ### Author Response · Authors · 2018-11-19
> **Subject: training the proposal distribution (4/5)**
>
>
> > Furthermore, to learn the proposed distribution, the paper applies the REINFORCE algorithm
> > under the proposed distribution q. But REINFORCE is known for large variance issue.
>
> This is a misunderstanding by the reviewer.  Following Lin & Eisner (2018), we use an interpolation of exclusive and inclusive KL divergence (equation (12)).
>
> REINFORCE corresponds to exclusive KL, which does have a variance issue.
>
> But in practice, our tuned interpolation coefficient placed *all* the weight on inclusive KL, which has no variance issue.  (This fact is reported as “beta=1” under equation (12).)  So our experiment effectively avoids REINFORCE altogether.  (Your comment may be the reason that beta=1 worked best for us, but see an alternative discussed below.  Note that Lin & Eisner found that beta < 1 worked best in their setting.)
>
> > Given that we already need lots of samples for the likelihood, it is unclear to me how
> > stable the algorithm could be in practice.
>
> We’re not sure what you mean here by “stable.”  Yes, we have a sampling-based method, but so do most people in the field right now!  As you know, stochastic gradient methods always make use of “lots of samples.”  Remember that SGD works because the errors average out to 0 over many stochastic gradient steps.  (If you don’t believe that, you should be rejecting all the deep learning papers that use SGD, right??)
>
> SGD methods succeed, both theoretically and practically, with even high-variance estimates of the batch gradient (e.g., where each stochastic estimate is derived from a *single* randomly chosen training example).  Thus, we should be fine with a noisy sampling-based gradient as long as it is *unbiased*.
>
> Our Monte Carlo integral estimates (taken from Mei & Eisner 2017, Appendix B.2) are in fact unbiased.  And as a result, our stochastic gradient estimate is also unbiased, as required (assuming that the observed complete data are distributed according to p).   Why?  Since beta=1, our stochastic gradient is simply (10).  No particle filtering or smoothing is used to estimate (10), because we train it using observed complete data, as explained in the last long paragraph of section 3.2.1.  The only randomness is the integral over [0,T] (similar to the one in (5)) that is required to estimate the term log q(z | x) in (10) … and as just noted, this integral estimate is unbiased.
>
> (It is true that if beta were < 1, we would compute the exclusive KL gradient using particle filtering or smoothing with M particles, and this would introduce bias in the gradient.  Nonetheless, since the bias vanishes as M goes to infinity, it would be possible to restore a theoretical convergence guarantee by increasing M at an appropriate rate as SGD proceeds -- see Spall (2003), p. 107.)
>
> As for whether the training algorithm could work “in practice” -- did you see the beautiful figure 2?  Our training method certainly appears to succeed “in practice.”  The trained proposal distribution is better on *virtually every example* in *12 different datasets*!  We as reviewers would be quite inclined to accept a paper with such clear results …
>
> Finally, recall that the paper has 3 algorithmic contributions: particle filtering, particle smoothing, and consensus decoding (as well as introducing a useful problem setting along with a well-thought-out evaluation metric).  Your question here about training the proposal distribution pertains only to particle smoothing.  Even if there were a problem here, the other contributions would stand.  But in fact, we see no problem here: we clearly demonstrate the value of training a proposal distribution.

---

> ### Author Response · Authors · 2018-11-19
> **Subject: efficiency of Monte Carlo methods (3/5)**
>
>
> > It turns out that the integral part of Equation 5 does not have an obvious analytical solution
> > under NHP. Then, we first need a set of  samples to approximate the likelihood evaluation.
>
> Well, this part of our method simply follows the algorithm given in the NHP paper (Mei & Eisner, NIPS 2017, sections B.2 and C.2), as we mention in our Appendix A (“integral computation”).
>
> Computationally it is not a problem.  A given run of particle smoothing begins by drawing O(I) time points from Uniform([0,T]), where I is the number of observed events.  All particles are evaluated using integrals that are estimated by evaluating the function at these time points.
>
> (Using the same time points for all particles gives a paired comparison that reduces the variance of the normalized importance weights.  Note also that because we sample time points uniformly, longer intervals between imputed events will tend to contain more points, which is appropriate.)
>
> > Later, we also need to sample particles.
>
> Our GPU implementation (which we will release) parallelizes the outer loop over particles.  We sample 50 particles in parallel in these experiments, but we have tested with 1000 particles in parallel as well.  So this is not a real problem with a GPU.
>
> > I am not quite convinced the computational efficiency of this approach in real applications of practice.
>
> We reported experiments that we performed to demonstrate the practicality.
> On average, drawing an ensemble of 50 particles takes
> 5s per example on the synthetic datasets (average length 15 events)
> 12s per example on the NYC taxi dataset (average length 32 events)
> 100s per example on the elevator dataset (average length 313 events)
>
> That is, 300-400 ms per event.  Such speeds are acceptable in many incomplete data applications, compared to the cost of collecting complete data.  Consider the applications on page 1 of the paper, all of which involve real-time decision making at a human timescale.
>
> > Also, there is no analysis either empirically or analytically about the impact of the
> > accumulative sampling errors on the inference performance.
>
> Mei and Eisner (2017, Appendix C.2) found that rather few samples could be used to estimate the integral: even sampling at only I time points gave a standard deviation of log-likelihood that was on the order of 0.1% of absolute (Mei, p.c.).
>
> What kind of “accumulative sampling errors” are you concerned about?  Remember that our integral estimate is *unbiased*, and the particle filtering estimate is at least consistent.  (Although it is true that the normalized particle weights are distorted both by the finite number of particles and the variance in the integral estimates, the variance of the integral estimation decreases---rapidly---as O(1/n) where n is the # of sampled time points.)

---

> ### Author Response · Authors · 2018-11-19
> **Subject: distributions other than NHP (2/5)**
>
>
> > The proposed technique is tightly connected to NHP … Can the proposed method also be applied to other processes?
>
> Thanks for the question. Yes, certainly.  The main technique is to use particle filtering or smoothing to sample from the posterior over complete sequences.
>
> Particle filtering is applicable to any temporal point process where the number of events is finite with probability 1, and where it is tractable to compute (or estimate) the log-likelihood of a prefix of a complete sequence.
>
> To extend this to particle smoothing, we developed a particular family of proposal distributions that is based on a continuous-time LSTM, as well as a method (Algorithm 1) for sampling proposals from such a distribution in the context of particle filtering.
>
> Our experiments use an NHP *model* of the complete sequence, together with a *proposal distribution* whose architecture happens to be almost identical to the NHP architecture (in mirror image, as it reads the future observed events from right to left).  However, *** our proposal distribution could also be used with other models ***: thus, we would also recommend it for particle smoothing of temporal point processes beyond just the NHP!
>
> The job of the proposal distribution is to get a good fit to the model’s complex posterior predictive distribution of the next event (which is defined by an integral over possible completions of the incompletely observed future).  A highly parameterized neural proposal distribution family like ours is designed to be flexible enough to do this, at least for non-pathological models.
>
> One caveat: Our proposal distribution does also take the state of the original point process into account.  In our case, that is the state of the NHP (h(t) in equation (9).  If you were using a different point process, you would need to replace h(t) with some other sufficient statistic of the history H(t).

---

> ### Author Response · Authors · 2018-11-19
> **Subject: minor misunderstanding (1/5)**
>
>
> > Experiments on synthetic datasets with 10 different initializations and two real datasets
>
> To be precise, it’s 10 completely different synthetic datasets.  Each dataset is drawn from a different distribution with randomly selected parameters.  Those distributions are not trained, so it’s odd to speak of “initializations.”

---

### Official Review · AnonReviewer2 · 2018-11-08
**This paper proposes an algorithm for missing data problem in continuous time events data (ie, point processes) where both past and future events are helpful.**

**Rating:** 5
**Confidence:** 4

**Review:**

This paper tackles a very important and practical problem in event stream planning. The problem is very interesting and the approach taken is standard.

The presentation of the paper is not clear enough. The notations and definitions and methods are presented in a complicated way. It's difficult to follows.

From the contribution point of view the paper looks like to be a combination of several existing and well developed approach: Neural Hawkes Process + particle smoothing + minimum bayes risk + alignment. It's not very surprising to see these elements together. It would have helped if the authors made it clear why each part is chosen and clearly state what is the novelty and contributed of the paper to the field.

The paper in its current format is not ready for publication. But it's a good paper and can be turned to a good paper for the next venue.

---

> ### Author Response · Authors · 2018-11-19
> **Subject: response to short late review**
>
>
> > It's difficult to follows.
>
> Thanks for acknowledging the importance of the problem.  We are sorry to hear that you found the paper too difficult to read in the limited time that you had available to review it.
>
> We worked quite hard on the exposition.  If you have specific suggestions that could reduce the difficulty, we will be happy to consider them for the camera-ready version.
>
> > But it's a good paper and can be turned to a good paper for the next venue.
>
> Thank you.  We agree that “it’s a good paper” already. :)  You provide few comments about how it could be “turned into a good paper for the next venue,” so we are not sure of your reasons for wanting to delay its publication.
>
> > It would have helped if the authors made it clear why each part is chosen and clearly state what is the novelty and contributed of the paper to the field.
>
> Yes, this is why we wrote section 6, “Discussion.”  Could you please reread that section?  It begins: “Our technical contribution is threefold,” and goes on to clearly describe each contribution and its novelty and importance.
>
> > several existing and well developed approach: Neural Hawkes Process + particle smoothing + minimum bayes risk + alignment
>
> Actually, we are further developing methods that are still in their infancy and are under current investigation.  NHP was first published in December 2017 and is being picked up by the community.  Neural methods for particle smoothing were first published in June 2018.
>
> Our alignment method required developing a new metric and alignment algorithm (section 4 and Appendix C, including Algorithm 2).  These are not groundbreaking but they did require some thought.
>
> Our MBR method required developing a new approximate search method (section 4 including Theorem 1, and Appendix D including Algorithm 3).
>
> We believe that the novel contributions of this paper are above threshold for publication in ICLR.  There is a lot of material in this paper.
>
> The paper also includes strong experimental results that should be of interest to the ML community and that demonstrate the potential of our methods for applied work.  We provided extensive pseudocode and will release our implementation.

---

### Meta-Review · Area_Chair1 · 2018-12-14
**Meta-Review for neural Hawkes particle smoothing paper**

**Confidence:** 5
**Recommendation:** Reject

**Metareview:**

All reviewers agree to reject. While there were many positive points to this work, reviewers believed that it was not yet ready for acceptance.